# Teaching Models to Teach Themselves : Reasoning at the Edge of Learnability

**Shobhita Sundaram** [* 1]  **John Quan** [2]  **Ariel Kwiatkowski** [2]  **Kartik Ahuja** [2]  **Yann Ollivier** [2]  **Julia Kempe** [2 3]

## Abstract

RL methods for scaling large reasoning models stall on datasets with low initial success rates, and thus little training signal. We investigate a fundamental question: *Can a pretrained LLM leverage latent knowledge to generate an automated curriculum for problems it cannot solve?* We explore this with SOAR: An asymmetric self-play framework that uses meta-RL to surface these pedagogical signals. A teacher model proposes synthetic problems for a student model, and is rewarded with its improvement on a subset of hard problems, thus grounding the curriculum in real student progress rather than intrinsic proxy rewards. Our study on the hardest subsets of math benchmarks (0/128 success) reveals three core findings. First, it is possible to realize bilevel meta-RL that unlocks learning under sparse, binary rewards by sharpening a latent capacity of pretrained models to generate useful problems. Second, grounded rewards outperform intrinsic learnability rewards used in prior LLM self-play, reliably avoiding typical instability and diversity collapse modes. Third, the structure and well-posedness of questions are more critical for learning progress than solution correctness. Our results suggest that the ability to generate useful stepping stones does not require the preexisting ability to solve the hard problems, paving a principled path to escape reasoning plateaus without additional curated data.

## 1. Introduction

Reinforcement learning with verifiable rewards (RLVR) has spurred an impressive rise in LLM reasoning capabilities (Guo et al., 2025; Team et al., 2025). However, this paradigm has a key limitation: *the model cannot learn from problems that it cannot already solve to some extent.* When problems are too difficult, sparse or non-existent rewards leave the model "stuck".

Past work has shown that curricula strongly affect generalization in RL (Bengio et al., 2009; Narvekar et al., 2020), with success in selecting "learnable" problems and adapting easy-to-hard progressions (Parashar et al., 2026; Chen et al., 2025b). Such curricula, however, require careful design and curated intermediate datasets (Kordi et al., 2026). Recent work exploits dense signals from test-case pass rates in coding problems (Sun et al., 2026) to address sparse rewards, but relies on curated test-cases.

Asymmetric self-play (Silver et al., 2018; Sukhbaatar et al., 2018; OpenAI et al., 2021) offers a potential solution to these limitations by enabling self-generated curricula. Here, we ask:

*Can a model break its reasoning plateau by generating its own stepping-stone curriculum?*

We posit that pretrained LLMs possess the capacity to directly generate a "stepping stone curriculum" to tackle hard problems. To investigate if this pedagogical signal is *present* and *extractable*, we design SOAR: an *asymmetric self-play framework* that uses hard problems as a guiding signal. Both the teacher and student are initialized from the target model; the teacher proposes questions-answer pairs that the student trains on with RL, and is rewarded with student improvement on a difficult subset. Critically, rather than using intrinsic rewards common to self-play, we use the difficult training dataset as a black-box grounding reward signal to guide the teacher towards useful questions for the student.

Intuitively, a pretrained model has already encountered a vast array of easy problems. Consider a difficult calculus question: While the model may be unable to directly answer correctly, it might still possess the latent knowledge to generate easy chain-rule exercises, without requiring a human-in-the-loop to identify and source such questions. We find that by leveraging pretraining knowledge, RL can effectively surface and amplify these latent pedagogical sig-

---
* Work done during an internship at Meta. [1]MIT [2]Meta FAIR [3]New York University. Correspondence to: Shobhita Sundaram <shobhita@mit.edu>.

*Proceedings of the 43rd International Conference on Machine Learning*, Seoul, South Korea. PMLR 306, 2026. Copyright 2026 by the author(s).

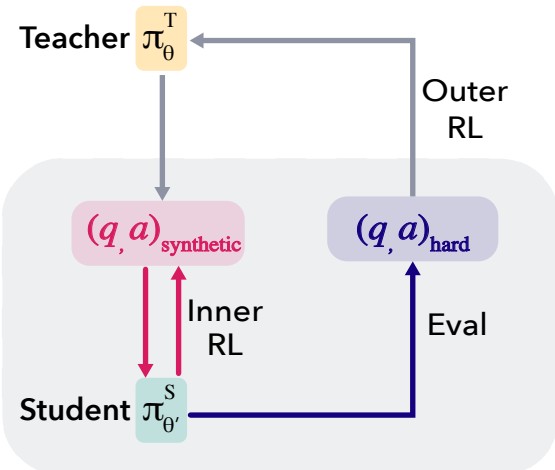

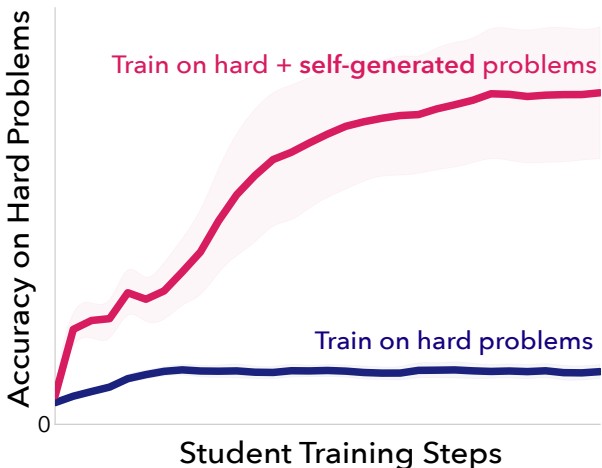

*Figure 1.* **Learning on hard problems by self-generating a curriculum.** We introduce **SOAR**: An asymmetric self-play framework that uses meta-RL to improve on difficult datasets where performance plateaus. **(left)** A teacher model generates synthetic problems for the student to train on with RL. The teacher is rewarded by the student's measured improvement on the real problems, providing a grounding signal. **(right)** Training on problems generated with SOAR (using hard problems as a teacher reward signal) outperforms direct training on the hard problems, allowing the student to break out of the performance plateau.

nals to generate useful question-answer pairs. Importantly, we do so without actually showing the model the hard questions; our framework recovers a useful curriculum just by using performance on the hard dataset as a reward signal.

Empirically, while direct training on the hard dataset fails, *the teacher in our framework learns to generate useful questions that get the student "unstuck" on the hard dataset, without actually seeing the hard problems*. Our main contributions, supported by an extensive multi-seed empirical study and ablations (over 600 runs), are the following:

- **Decoupled teaching and solving:** A model's ability to generate effective "stepping stones" for hard problems is distinct from its ability to solve them. Self-play can exploit this asymmetry to generate problems that *expand the learning frontier*, enabling progress where direct RL fails. While the base model has the capacity to propose useful questions, meta-RL sharpens this noisy distribution into a reliable learning signal.

- **Proof-of-concept of self-generated curricula** with SOAR (**S**elf-**O**ptimization via **A**symmetric **R**L), an asymmetric self-play framework that rewards the teacher for student progress on hard problems. On hard subsets of MATH and HARP, self-generated problems improve performance (e.g., 4× pass@1 and 2× pass@32 on MATH, 2× pass@1 and 1.5× pass@32 on HARP). These problems also transfer to unlock learning on datasets that they were not optimized for.

- **Mitigating self-play collapse with grounded rewards:** Grounding teacher rewards in student progress on real problems improves performance over intrinsic rewards common in self-play, which are prone to

instability and collapse of question diversity.

- **Question structure over solution correctness**: Problem structure and difficulty calibration matter more for escaping plateaus than answer correctness; generated questions provide useful gradient signal even when the majority of answers are incorrect.

These results, backed by a comprehensive empirical study, show that grounded meta-RL can escape genuine learning plateaus by letting models discover for themselves what data they need to learn from to expand their learning frontier.

**Conflicts of Interest Disclosure.** The authors JQ, AK, KA, YO, JK were employed by Meta FAIR at the time of writing this paper, which leads the development of the Llama model series, which was evaluated in this paper.

## 2. Related Work

For an extended review of related literature see Section A:

**Self-Play and Teacher-Student Setups.** Self-play aims to achieve autonomous capability growth, exemplified by game-playing agents such as AlphaZero (Silver et al., 2018), and asymmetric teacher-student setups that induce automatic curricula (Sukhbaatar et al., 2018; OpenAI et al., 2021). LLM self-play methods encounter specific challenges: language rewards are extremely sparse and brittle. For mathematical problems, correctness is essentially binary and offers no gradient toward partial solutions. Thus, essentially all modern methods optimize for self-consistency or solution quality. Earlier works (Chen et al., 2024; Wang et al., 2025; Singh et al., 2024; Ye et al., 2024) presuppose the existence of well-formed input prompts or curated

high-quality questions.

A series of near-contemporary works leverages pretrained LLMs as question generators to create "fully data-free" co-evolving systems (Zhao et al., 2025a; Huang et al., 2026; Kuba et al., 2025; Fang et al., 2025; Chen et al., 2025a). These works leverage intrinsic or proxy rewards such as majority vote, learnability, reward-model preferences, or gradient magnitudes. Due to optimizing intrinsic or proxy objectives, they risk drifting to degenerate or unlearnable tasks, are sensitive to reward hacking and lack progress guarantees (Chae et al., 2025). Prolonged RL with self-rewards often causes sudden and complete performance collapse (Shafayat et al., 2025; Chae et al., 2025), when rewards vanish or when generator and solver objectives misalign, especially in discrete, symbolic domains with essentially binary correctness signals. This fragility raises the broader question of whether self-improvement driven by intrinsic or self-generated rewards can be sustained within RL. To our knowledge, our work is the first for LLM self-play to ground the curriculum generation in a concrete failure regime instead of internal difficulty proxies.

**Curriculum Learning in RL.** Automated curriculum design has a long history (Bengio et al., 2009; Graves et al., 2017; Narvekar et al., 2020; Parashar et al., 2026) focusing on *reordering* or *selecting* existing data to enable or accelerate learning, or, in the context of RL, to help agents acquire complex behaviors by first mastering simpler tasks. For LLM training, curricula are applied over curated prompts or problem categories, using proxy signals such as gradient norms or advantage/difficulty estimates to guide selection (Team et al., 2025; Dennis et al., 2020; Wen et al., 2025; Yu et al., 2025; Bae et al., 2026; Chen et al., 2025b; Jiang et al., 2025). By contrast, our goal is not to arrange data but to *self-generate tasks* to elicit learning on a fixed, verifiable hard dataset where standard RLVR fails.

Another line of work instead generates a distribution of environments suitable for agent capabilities (Dennis et al., 2020; Racaniere et al., 2020; Jiang et al., 2020; 2021). These works find that unconstrained objectives (e.g., minimax adversarial objectives) lead to degenerate curricula, instead optimizing regret or learning potential. We diverge by grounding rewards in student progress on a fixed target set.

**Intrinsic Rewards versus Bilevel Optimization.** Various intrinsic rewards have been studied across robotics, simulation, and task-agnostic settings for curricula generation and exploration (Schmidhuber, 1991; Pathak et al., 2017; Colas et al., 2019; Blaes et al., 2019; Colas et al., 2022; Sancaktar et al., 2023). In self-play, the use of proxy rewards is often not merely a design preference but a pragmatic simplification. It avoids facing an explicit bilevel optimization problem: an appealing but challenging objective where the output of one optimization (in this instance the optimization of the student trained with RLVR on the teacher's question-answer pairs) is fed into another optimization loop (the performance improvement of the student on the hard dataset). Such bilevel optimization appears in meta-learning (Finn et al., 2017; Nichol et al., 2018), hyper-parameter learning (Maclaurin et al., 2015) and—partially inspiring our work—in dataset distillation, where an outer loop optimizes a small dataset for an inner training loop to achieve good target performance (Wang et al., 2018; Deng & Russakovsky, 2022; Feng et al., 2024). In general, such approaches become intractable, as the inner loop involves a multi-step computation with a large number of steps, requiring backpropagation through time (BPTT), unrolling the inner loop and taking meta-gradients. Our approach avoids the need to unroll the inner loop thanks to the use of RLOO in the outer loop, using the performance improvement of the student as the reward to reinforce question-answer sets. This is the first instance of "double meta-RL loop" we are aware of in the context of self-play for LLMs.

## 3. Method

Can a pretrained LLM leverage latent knowledge to generate question-answer pairs for problems it cannot solve? And can this be achieved in domains with sparse, binary rewards lacking automatic question verification? To explore this, we introduce SOAR: an asymmetric self-play framework that uses meta-RL to surface such pedagogical signals. Critically, SOAR grounds the teacher reward in measured student progress rather than intrinsic proxy rewards. If the model can generate useful stepping stones despite being unable to solve the original problems, this would suggest that the latent knowledge exists, and is extractable without human curation.

Let $\pi_\theta$ be a language model with parameters $\theta$. We assume access to a dataset $\mathcal{D} = \{(q_i, a_i)\}_{i=1}^{|\mathcal{D}|}$ of *difficult* question-answer pairs ($\pi_\theta$ produces 0/128 successful generations). $\mathcal{D}$ is split into train and test sets: $\mathcal{D}_{train}$, $\mathcal{D}_{test}$. To improve the performance of $\pi_\theta$ on $\mathcal{D}_{test}$, the natural approach is to train $\pi_\theta$ directly on $\mathcal{D}_{train}$ using RL (*e.g.*, REINFORCE, GRPO, RLOO, etc). However, for difficult datasets, this may not improve performance due to the sparsity of positive rewards. We instead use this "failure regime" as a testbed to see if the model can autonomously recover intermediate problems that make these hard problems more learnable.

### 3.1. Overview

Our framework adopts the teacher-student setup of asymmetric self-play, to "kickstart" learning on datasets where the initial success rate is too low for successful training. We instantiate two copies of the same model: a teacher $\pi_\phi^T$ and a student $\pi_\theta^S$. At step zero, $\theta = \phi = \theta_{base}$.

The teacher's role is to generate synthetic problems that provide the student with the necessary gradient signal to escape the performance plateau. Intuitively, while the teacher may be unable to solve a difficult problem directly, it may still possess the knowledge to *generate* easier problems that provide a non-zero reward to the student and shift its policy towards progress on the original problem.

We formulate this problem as a bilevel optimization problem. The objective is to generate a small synthetic dataset $\mathcal{X} = \{(q_i, a_i)\}_{i=1}^{n}$ of question-answer pairs such that training $\pi_\theta^S$ on $\mathcal{X}$ with RL improves performance on the target domain.

$$\max_\phi \quad \mathbb{E}_{\mathcal{X} \sim \pi_\phi^T} \left[ R\left( \pi_{\theta'(\mathcal{X})}^S, \mathcal{D}_{train} \right) \right]$$
$$\text{subject to} \quad \theta'(\mathcal{X}) = \text{RL-UPDATE}(\theta, \mathcal{X}), \quad (1)$$

where RL-UPDATE describes the RL training procedure of the student on $\mathcal{X}$, yielding parameters $\theta'(\mathcal{X})$, and $R$ denotes the updated student's performance on $\mathcal{D}_{train}$.

Such bilevel objectives have strong precedence in meta-learning (Finn et al., 2017; Nichol et al., 2018), hyperparameter learning (Maclaurin et al., 2015) and dataset distillation (Wang et al., 2018). In general, such approaches become intractable, requiring "backpropagation through gradient descent". To avoid the associated computational difficulties, we instantiate objective (1) as a nested meta-RL loop:

- **Outer RL loop**: Train the **teacher** with RLOO (Ahmadian et al., 2024) to generate question-answer pairs.

- **Inner RL loop**: Train the **student** with standard RLVR (also RLOO) on teacher-generated problems. The subsequent performance improvement of the student on $\mathcal{D}_{train}$ is the black-box reward signal for the teacher.

We do not assume automatic verification of synthetic question well-posedness or answer correctness (as *e.g.,* in coding tasks in Zhao et al. (2025a)). Instead, the teacher generates both the question and answer, treating the question utility as an emergent property of the teacher's reward signal. Critically, we ground the teacher's objective in measured student progress on $\mathcal{D}_{train}$, rather than intrinsic proxies such as learnability, as done in prior work. SOAR only rewards a synthetic question-answer pair if training on it improves student performance on ground-truth problems. This *black-box grounding signal* tethers question generation to real learning progress, implicitly penalizing degenerate problems and reward hacking. The teacher is not shown the hard problems during training, but rather discovers useful stepping stones purely from this student improvement signal.

See Algorithm 1, illustrated in Figure 2, for the full method.

### 3.2. Outer Loop: Teacher Training

We train the teacher with RLOO to generate problems that demonstrably improve student performance. Let $g$ denote the RLOO group size and $n$ the size of the generated dataset $\mathcal{X}$. At each iteration, we sample $g \cdot n$ rollouts $y_1, \ldots, y_{gn}$ from $\pi_\phi^T$, subdivided into $g$ datasets of $n$ items each: $\mathcal{X}_1 = \{y_1, \ldots, y_n\}, \ldots, \mathcal{X}_g = \{y_{g(n-1)}, \ldots, y_{gn}\}$. Since we cannot automatically verify the answers to proposed problems, we prompt the teacher to generate both the question *and* answer. Each rollout $y_i$ is parsed into $y_i = (q_i, a_i)$ (described in Appendix B.2; we may need to sample multiple times to obtain a parseable $y_i$).

Each dataset $\mathcal{X}_k$ receives a reward. At each outer-loop iteration we subsample a set of *reward questions* $\mathcal{Q}_R \sim \mathcal{D}_{train}$ from the original training set. We train the student on each dataset $\mathcal{X}_k$, for a fixed number of steps, resulting in trained student $\pi_{\theta'_k}^S$. The dataset-level reward $R(\mathcal{X}_k)$ is then the average greedy success of $\pi_{\theta'_k}^S$ on $\mathcal{Q}_R$ relative to the success of a baseline student model $\pi_\theta^S$:

$$\mathcal{R}(\mathcal{X}_k) = \text{Acc}(\pi_{\theta'_k}^S(\mathcal{Q}_R)) - \text{Acc}(\pi_\theta^S(\mathcal{Q}_R)),$$

where $\pi_\theta^S$ is the initial student when starting the inner loop. We subtract the initial student accuracy so that teacher rewards are normalized across outer-loop steps, necessary for the student promotion mechanism introduced in Section 3.3.

To mitigate reward variance, we average rewards over $r$ parallel student trainings per dataset. This averaged reward is assigned to each rollout in $\mathcal{X}_k$ to update the teacher.

### 3.3. Inner Loop: Student Training

The student $\pi_\theta^S$ trains on the teacher-generated dataset $\mathcal{X}_k$ using RLOO for 10 steps (batch size 8), long enough to induce measurable movement while minimizing computational cost. After each inner loop the student reverts to the baseline policy for the next iteration.

A key question is whether the teacher can adapt to student improvement. To address this, we introduce a *promotion* mechanism to accumulate student improvement and useful questions across inner loops. We track a rolling moving average of teacher rewards $\bar{R}_t$; when it exceeds a fixed threshold $\tau$, we update the baseline student $\pi_\theta^S$ to the student trained on the best $\mathcal{X}_k$. Subsequent rewards measure improvement relative to this new baseline (Appendix B.3). We denote the accumulated datasets that led to student promotions as $\mathcal{D}_{best}$; these constitute the Promotion Questions (PQ) that we evaluate in our experiments.

## 4. Experiment Setup

### 4.1. Models and Datasets

Our experiments mainly use `Llama-3.2-3B-Instruct`, with ablations extending to `Llama-3.1-8B-Instruct` in Appendix

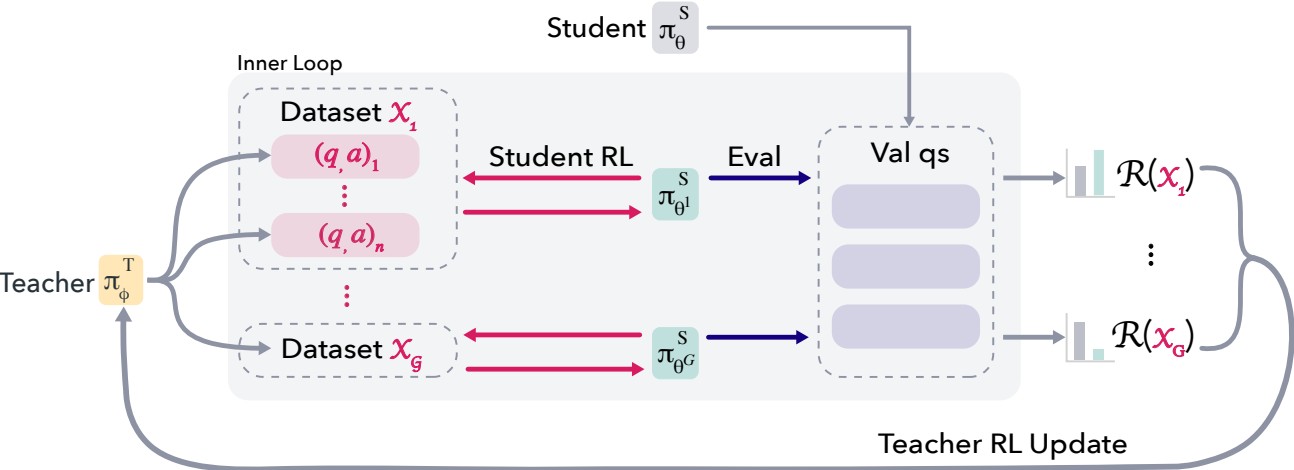

*Figure 2.* **The SOAR meta-RL Loop.** The teacher and student are initialized from the same model. In the **outer RL loop** the teacher generates candidate question-answer pairs that are partitioned into datasets. In the **inner RL loop**, the student is trained for 10 steps on the candidate problems and evaluated on sampled hard problems. The teacher is rewarded based on the resulting student improvement over the student baseline, grounding the synthetic curriculum in real learning progress.

D.4. We focus on math reasoning benchmarks to study the prototypical setting of sparse, binary rewards, without automatic question-answer verification (as in code): MATH (Hendrycks et al., 2021), HARP (Yue et al., 2024), and OlympiadBench (He et al., 2024). For each dataset, we sample 128 times per problem with the target model, retaining those with a 0/128 success rate. We call these subsets *fail@128* datasets; 128 serves as a practical but stringent threshold at which, empirically, direct training yields only marginal improvement. Each is randomly split 50-50 into training and test sets. Given the low baseline pass rates on fail@128 problems, this larger test set is necessary to distinguish performance gains from stochastic variance. Details in Appendix B.5.

### 4.2. Teacher-Student Training

We initialize the teacher and student from the base model and train SOAR on MATH and HARP, keeping Olympiad-Bench held-out. We allocate 200 outer-loop steps based on compute constraints. Each outer-loop iteration samples $n = 64$ problems ($\mathcal{X}$) from the teacher, and 64 reward questions ($\mathcal{Q}_R$) from the fail@128 train set ($\mathcal{D}_{train}$). We promote the student baseline when the 3-step moving average of teacher rewards exceeds $\tau = 0.01$. Full hyperparameters are in Appendix B.7, a sensitivity analysis for $\tau$ and $n$ in Appendix D.2, and SOAR training dynamics in Appendix E.

### 4.3. Evaluation

Once training completes, we test if the generated problems improve $\mathcal{D}_{test}$ performance. Based on observations of teacher reward plateaus in initial runs, we evaluate the teacher where training rewards stabilize: step 200 for MATH

and step 170 for HARP. We assess two aspects of SOAR:

**Promoted Student (PS).** For training runs that reached multiple promotions, we evaluate the student model with the best validation performance (*i.e.,* best $\mathcal{D}_{train}$ greedy accuracy) on the test set to measure direct performance gains from SOAR. In practice we observe a maximum of four promotions; thus the PS model has been trained on one of {128, 192, 256} synthetic questions.

**Promotion Questions (PQ).** We train a fresh base student with standard RLOO on a combination of PQ ($\mathcal{D}_{best}$) and the fail@128 train set. This isolates the value of the synthetic questions, separate from the specific training trajectory of the promoted student. We denote PQ from MATH and HARP training as PQ-MATH and PQ-HARP. In Appendix B.6 we discuss how we mix synthetic and real data.

We compare to the following baselines:

***Hard-Only.*** We train directly on the $\mathcal{D}_{train}$ (real fail@128 train set) with a standard group size of 32. To disentangle the effects of the meta-RL loop from just using additional compute, we also train with group size 128 on MATH.

**Intrinsic Teacher (*Intrinsic-T*).** To isolate the effects of grounding rewards, we compare to an intrinsic, data-free baseline. We train using the same procedure and hyperparameters as SOAR, but replace the grounded signal with a learnability objective (Zhao et al., 2025a; Sukhbaatar et al., 2018) that rewards questions of moderate difficulty. We evaluate by training a fresh student on 128 problems sampled from this teacher (*Intrinsic-T*) alongside the fail@128 train set, following the PQ evaluation protocol. Learnability details in Appendix B.4.

**SeRL (Fang et al., 2025).** SeRL serves as contem-

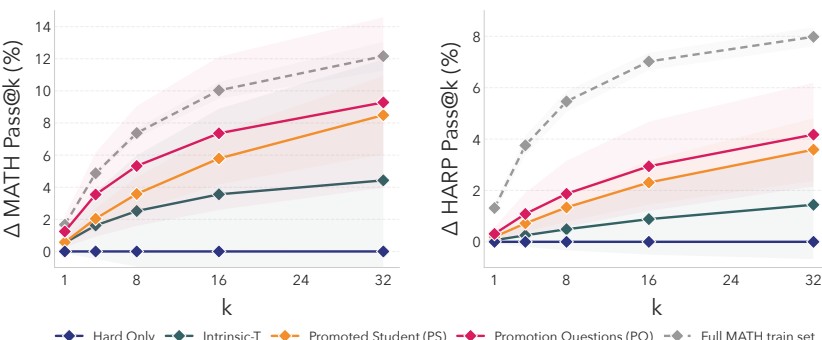 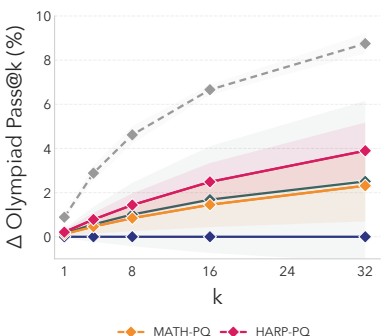

*Figure 3.* **Performance on MATH/HARP fail@128 (improvement over *Hard-Only*).** Synthetic problems generated with SOAR (PQ) and inference with the promoted student (PS) outperform direct training on fail@128 train sets (*Hard-Only*), and sampling from teachers trained with intrinsic rewards (*Intrinsic-T*). For reference, *Hard-Only* MATH pass@$k$ for $k \in \{1, 4, 8, 16, 32\}$ is $\{0.5, 1.7, 3.2, 5.7, 9.6\}$. Full trajectories in Figure 9; absolute performance and further evaluations in Tables 4-5. Shaded regions are $\pm$ 1 SD over 6-12 seeds (Appendix B.8).

*Figure 4.* **Transfer performance to OlympiadBench fail@128 subset (improvement over *Hard-Only*).** Questions optimized for MATH and HARP transfer to a held-out dataset. Absolute performance, including PS evaluation, is in Table 6.

porary self-play baseline that self-evolves a curriculum from a seed set using self-rewards and diversity/learnability filters. We train SeRL with the MATH and HARP fail@128 train sets as the seed sets, until convergence. We use the hyperparameters specified by the authors for Llama-3.2-3B-Instruct.

**Upper bound.** We train a fresh student the full MATH train split (6750 problems) plus the fail@128 train set. This serves as an upper bound reference for performance with human-curated stepping stones.

**Metrics.** We report the pass@k accuracy on the held-out fail@128 test set for $k \in \{1, 4, 8, 16, 32\}$, using 32 samples per problem. We report mean and standard deviation over 6-12 seeds, nested across teacher/student training (Appendix B.8). For evaluations with fresh students, we do early stopping based on training reward convergence due to our small dataset size and differing convergence rates (Appendix B.6).

## 5. Results

### 5.1. Meta-RL Discovers Effective Questions.

While curriculum learning is well-studied in RL, it is not obvious that synthetic questions can help a model move "beyond sharpening" its existing distributions. We show that self-generated stepping stones provide a learnable gradient that unlocks improvement in stalled regimes. This occurs without the teacher seeing the target problems; instead, meta-RL sharpens the teacher's policy, discovering useful curricula solely by optimizing for student progress.

**PQ kickstarts learning on hard subsets.** Both PS and PQ substantially outperform *Hard-Only* and *Intrinsic-T* baselines, with larger gains at higher $k$. Figure 3 shows *im-*

*provement over Hard-Only*. *Hard-Only* test trajectories are in Figures 5; all absolute numbers and trajectories are in Appendix C.1-C.2. Inference with the base model achieves non-zero pass@$k$ due to stochastic sampling with different seeds than were used for the initial fail@128 filtering; nonetheless, *Hard-Only* training plateaus.

Inference with PS achieves +8.5% pass@32 on fail@128-MATH and +3.6% pass@32 on fail@128-HARP over *Hard-Only*. PQ achieves higher mean performance (+9.3% pass@32 on MATH, +4.2% on HARP), indicating that *the synthetic questions, rather than a fortunate student training trajectory, drive the performance gains.* Both *Intrinsic-T* and SeRL (Tables 4-5) perform worse across datasets, validating that *grounded rewards are needed to discover the right questions.*

Synthetic questions also shift the student policy to make previously hard problems learnable; student learning curves on MATH show continued improvement after transitioning to real fail@128 training (Figure 9). These effects significantly outstrip what can be achieved from repeated sampling alone on fail@128 data. *Hard-Only* with a group size of 128 (4× extra compute) achieves only +2.8% pass@32 (Table 4); furthermore, extending *Hard-Only* training from 1500 to 6500 steps does not improve performance (Figure 10).

**OOD generalization.** Figure 4 shows that synthetic questions from PQ-MATH, PQ-HARP, and *Intrinsic-T transfer* to OlympiadBench, an OOD dataset. Cross-dataset transfer, despite no OOD optimization, suggests that synthetic curricula can capture generalizable reasoning pathways.

**Oracle comparison to real curated data.** Our regime assumes that we only have access to hard problems, to study the case where additional expert-curated data is not available or not known. As a strong upper-bound, we compare to the

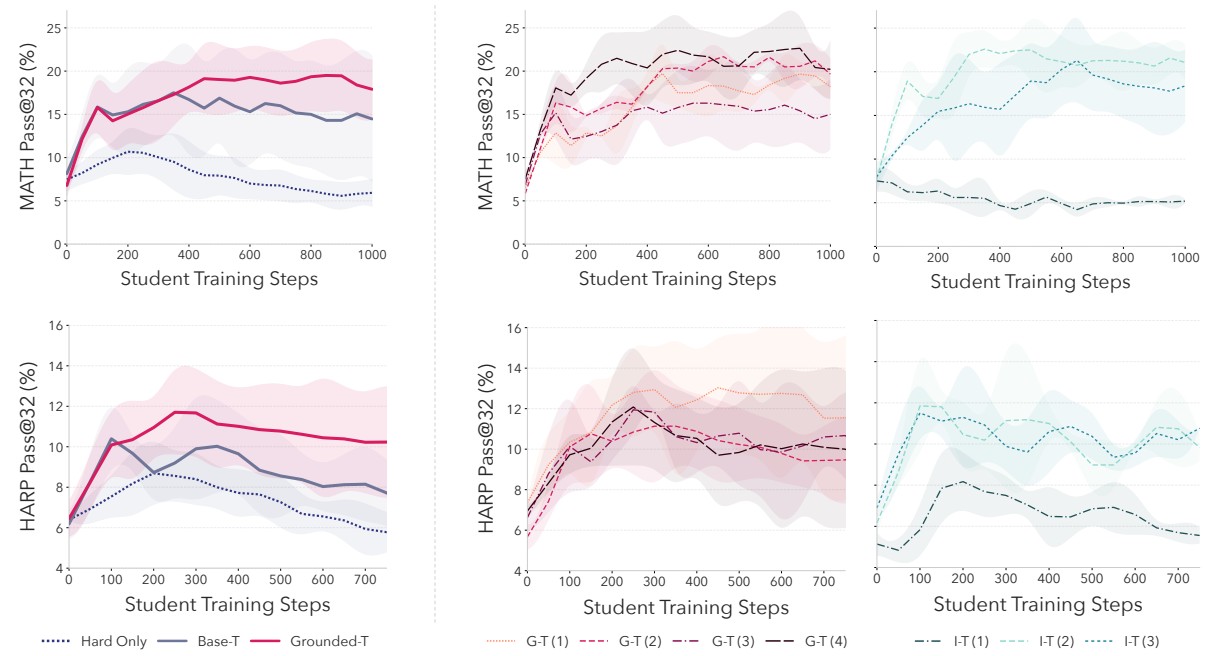

*Figure 5.* **Grounded rewards lead to more stable teacher policies.** We evaluate trained teacher policies by sampling questions and training fresh students. **(Left)** Test pass@32 comparison between students trained with questions sampled from *Grounded-T* and *Base-T* (*Hard-Only* also shown for reference). *Grounded-T* outperforms *Base-T* and exhibits more stable student trajectories. **(Right)** Pass@32 trajectories for fresh students trained with individual *Grounded-T* teacher seeds (red) and *Intrinsic-T* teacher seeds (green). Questions from *Grounded-T* yield consistent student trajectories, whereas *Intrinsic-T* exhibits higher variance across teachers, including a failure mode where I-T (1) causes student collapse. Shading shows ±1 SD. Curves for other pass@k and OlympiadBench are in Figures 11-13.

"oracle" case where curated extra data is available. We train students on fail@128 + the full official MATH training set (6750 problems) as a representative pool of abundant, easier questions. We also compare to training with 128 random MATH/HARP questions in Appendix C.2, which performs similarly to training with the full dataset. Synthetic PQ-MATH questions recover 75% of the performance gains from full-MATH training, and PQ-HARP recover 50%. Notably, HARP-PQ (128/192 questions) outperforms 128 real HARP questions, and matches 128 real MATH questions.

Direct inference on fail@128 test problems with the final *trained teacher policy* model does not improve over base model performance (Appendix C.2), indicating that generator and solver abilities are largely independent.

> **Takeaway:** A model's *pedagogical* ability can be decoupled from its *task-solving* ability. Grounded meta-RL expands the "learnability frontier" by surfacing synthetic questions that enable improving over reasoning plateaus.

### 5.2. Grounded Rewards Yield Stable, Diverse Teachers.

While the main utility of SOAR is in surfacing a set of useful teacher-generated questions (PQ), we now shift focus to the trained teacher policies themselves. We perform a controlled study of teacher objectives to probe the effects

of meta-RL, and show that grounded rewards (as in SOAR), versus intrinsic ones, yield stronger teacher policies. We evaluate teachers trained with grounded rewards (*Grounded-T*), intrinsic rewards (*Intrinsic-T*) and the base model (*Base-T*) by sampling question-answer pairs from these policies and training fresh students. In Appendix C.3 we also ablate SOAR without the student-promotion mechanism, to validate its necessity.

We evaluate four *Grounded-T* seeds per dataset to cover a range of final promotion stages and three *Intrinsic-T* teacher seeds. We sample 128 questions from each teacher and train 2-3 fresh students on the synthetic questions and real fail@128 train set ($\geq 9$ student runs per metric).

**The teacher policy generates useful questions.** Student test performance curves in Figure 5 reveal that questions sampled from *Grounded-T* improve over *Hard-Only*. Results are competitive with PQ on MATH and HARP, validating that the useful pedagogical signal is not just captured in the set of evolved questions, but is also learned by the teacher policy. Further ablations show that sampling larger datasets from *Grounded-T* reduces the variance of student outcomes (Appendix D.1) and that the student-promotion mechanism improves the teacher policy (Appendix C.3).

**Meta-RL sharpens the question distribution.** In Figure

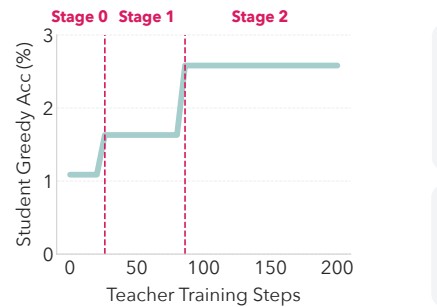

### Stage 1

**Q:** Tom has a rectangular garden with a length of 15 meters and a width of 8 meters. The cost of buying a tarp to cover the garden is 10.50 per square meter. If Tom buys more than one tarp, what is the least cost he can expect to pay?
**A:** 1260

**Q:** What is the area of the path that is added around a rectangular garden with dimensions 15 meters by 8 meters when a 1-meter-wide garden path is built?
**A:** 50

### Stage 2

**Q:** Given the polynomial f(x) = x³ - 6x² + 11x - 6, factorize and solve for its roots.
**A:** (2, 3, 1)

**Q:** Find the dynamics of the function f(x) = 2sin²(x+3π/2) + 3cos²(x-2π/5), where 0 ≤ x ≤ 2π.
**A:** -cos²(x + 3π/2) + 3sin²(x + 3π/2)

**Q:** What is the value of x in the equation: x³-y³ = (2x + 3y)y²?
**A:** 3

*Figure 6.* **Qualitative evolution of generated questions.** (Left) Baseline student performance during a SOAR run on HARP. The y-axis shows greedy accuracy on the *fail@128 train set* over promotion stages. (Right) Sampled teacher questions at different promotion points. Content and style shift from word problems and basic formulas (stage 1) to concise, equation-heavy problems (stage 2). Many effective "stepping stones" include incorrect solutions, suggesting that structural and conceptual content provide sufficient learning signal.

5 (left) we overlay student training curves for *Grounded-T* questions and *Base-T* questions. *Grounded-T* students consistently track the upper envelope of *Base-T* performance for MATH/HARP, with lower variance on MATH. The existence of successful runs from *Base-T* reveals the ability to generate useful stepping stone questions is latent in the model; meta-RL improves *Grounded-T* by *sharpening the teacher* to output questions that more reliably provide useful gradient signal. This is yet another example of the sharpening mechanism of RL (Yue et al., 2025; Zhao et al., 2025b; Tsilivis et al., 2026), but here leveraged for curricula. On OlympiadBench, where the target distribution differs substantially from the teacher's training signal, *Grounded-T* and *Base-T* learning curves overlap more (though *Grounded-T*-HARP achieves best performance), suggesting that meta-RL primarily sharpens in-domain pedagogical signals.

**Fragility of intrinsic proxies.** Figure 5 (right) compares aggregate student training curves for individual *Grounded-T* and *Intrinsic-T* teacher seeds. Students trained with questions from different *Grounded-T* seeds exhibit highly similar trajectories, indicating that grounded rewards lead to stable teacher policies. *Intrinsic-T* teachers produce, on average, worse and more volatile outcomes, with a clear separation in performance between students trained with different *Intrinsic-T* seeds across MATH, HARP, and Olympiad-Bench. While some *Intrinsic-T* teachers produce highly effective curricula, the objective is subject to a high-variance failure mode: one out of three teacher seeds exhibits collapse across all datasets, yielding little or no progress on the target problems. This reinforces observations from the literature that RL with self-rewards is prone to reward hacking, or the decoupling of the intrinsic reward from actual task mastery (Shafayat et al., 2025; Chae et al., 2025).

**Grounded training sustains diversity.** To probe how meta-RL shapes the teacher's generative distribution, in Table 1 we measure the semantic diversity of datasets from different teachers with the Vendi Score ($VS$) (Friedman & Di-

eng, 2023) using `Qwen3-8B` embeddings (Zhang et al., 2025). *Grounded-T* (MATH) and *Grounded-T* (HARP) match the diversity of *Base-T* ($VS = 34.91$), with PQ showing only a small decline from the base model ($VS = 31.75$). In contrast, *Intrinsic-T* collapses into a narrow conceptual space ($VS = 10.82$), providing evidence of reward-hacking and an explanation for the observed instability. This suggests that grounded rewards successfully avoid the diversity collapse often seen in RL-loops (Song et al., 2025), while intrinsic rewards fall prey to it. Indeed, we also observe a decline in the diversity of teacher completions during meta-RL with learnability rewards (Appendix E).

*Table 1.* **Semantic diversity of synthetic datasets**. Diversity is measured with the Vendi Score ($VS$), representing the effective number of unique semantic concepts. All metrics are standardized to 128 questions via bootstrap subsampling ($k = 100$ iterations). Our grounded reward (*Grounded-T*) preserves model diversity better than intrinsic rewards (*Intrinsic-T*).

| Method | Vendi Score ($VS$) | Std. Dev ($\sigma$) |
|---|---|---|
| *Base-T* | **34.91** | **1.74** |
| *Grounded-T* (HARP) | 34.66 | 1.74 |
| *Grounded-T* (MATH) | 31.99 | 1.54 |
| PQ | 28.33 | 1.55 |
| *Intrinsic-T* | 10.82 | 1.01 |

> **Takeaway:** Effective questions are latent in the base model, but hard to find. Grounding rewards in student progress "sharpens" the teacher's noisy distribution of questions into a stable, diversity-preserving policy, whereas intrinsic rewards are prone to instability and diversity collapse.

## 5.3. Question Structure over Answer Correctness.

While conventional wisdom suggests that question-answer correctness is most important, our results suggest that the *conceptual content and structure of questions* is more important for models on learning plateaus.

Figure 6 shows qualitative examples of PQ questions at different stages of a sample `SOAR` training trajectory, exhibiting shifts in style and conceptual focus as the baseline student improves. We annotate synthetic questions with `Claude-4.5-Sonnet` as an oracle judge, and observe that only 32.8% of PQ problems contain a fully correct solution, while 63% are considered mathematically well-posed (Appendix C.4). This suggests that for models stalled on a performance plateau, structural and contextual cues of a question are more important for kickstarting learning than a correct answer. Indeed, *Intrinsic-T* questions have *higher* correctness (55%) but perform worse, likely because of lack of diversity (Section 5.2). A more detailed taxonomy of error types is in Appendix C.4. Meta-RL decreases question ambiguity errors relative to *Base-T*, validating the importance of question coherence over answer correctness.

To better isolate these effects, we compare training on the subset of HARP-PQ questions with correct answers, to training on the well-posed subset (both correct and incorrect answers) in Appendix C.5. Performance improves with the addition of well-posed questions with incorrect answers. Our experiments with *Base-T* (which, like *Grounded-T* and *Intrinsic-T*, is filtered for correctly formatted questions) show that question format alone is not behind these results.

> **Takeaway:** For models at learning plateaus, problems that have conceptually diverse and coherent *questions* can provide useful gradient signal even without having precisely correct *answers*.

# 6. Discussion and Conclusions

**Breaking the sparse-reward plateau in RL fine-tuning.** Our work establishes a way to kickstart RL fine-tuning when the initial success rate is too low to collect RLVR signal. Generating and training on question-answer pairs (even if not correct), with the right meta-RL self-play loop, can be enough to provide nonzero signal on the original hard problems. Our setup shows that *generating stepping-stone questions to solve a problem does not require the preexisting ability to solve that problem*, and that *meta-RL sharpens this latent ability in the pretraining distribution*. A central contribution is that we show how to make this grounded bilevel meta-RL loop work in practice. The gap in performance shows the importance of this point.

**Grounded rewards mitigate self-play collapse.** The intuition of a gap between the abilities to generate and solve problems lies at the core of self-play. However, we show that it is crucial to go beyond pure curiosity by grounding in actual performance. Prior LLM self-play approaches use purely intrinsic rewards, such as learnability and self-consistency, to train the teacher. We find, however, that these objectives are prone to reward hacking, diversity collapse, and instability across seeds because they are decoupled from task performance. Grounding the teacher reward in measured student progress on real problems prevents teacher degeneration and preserves diversity.

**Expanding the learning frontier.** Our results tie to the broader debate on whether RL fine-tuning truly expands a model's learning frontier, or merely sharpens latent abilities (Yue et al., 2025; Zhao et al., 2025b; Tsilivis et al., 2026). Our work indicates that meta-RL can expand the envelope of learnability beyond what direct RLVR can achieve. As a "North Star" thought experiment, consider a future model trained on the entire mathematical literature: a proof of a Millennium Problem such as the Riemann Hypothesis may already be latent in pretraining, yet successful learning would hinge on recovering the right sequence of intermediate lemmas and theorems that make the proof *learnable* to a student reasoner. In this view, just as RL is believed to amplify useful subsets of pretraining data, meta-RL could retrieve the stepping-stone question–answer pairs embedded in the teacher's vast training corpus. We believe our results provide concrete evidence that a moderate amount of grounded meta-RL can elicit such capabilities that remain inaccessible through repeated sampling alone.

**Limitations.** Our primary limitation is the computational cost of running bilevel RL loops (Appendix B.9). While inner loop training is relatively cheap (10-20 steps) it necessitates training parallel students for stability. Our ablations in Table 4 and Figure 10 show that reallocating extra compute to direct training on hard problems does not recover the gains from meta-RL. Our work is a proof of concept for grounded rewards in this setting; more efficient reward proxies and scaling beyond 3-8B models are rich avenues for further work.

## Acknowledgements

We thank Cansu Sancaktar, Reyhane Askari Hemmat, and Phillip Isola for helpful discussions. JK thanks the Simons Foundation for support through the Collaborative Grant "The Physics of Learning and Neural Computation". This work was supported by an NSF GRFP fellowship to SS. This work was also supported under project ID 43 as part of the Swiss AI Initiative, through a grant from the ETH Domain and computational resources provided by the Swiss National Supercomputing Centre (CSCS) under the Alps infrastructure

## Impact Statement

This paper presents work whose goal is to advance the field of Machine Learning. There are many potential societal consequences of our work, none which we feel must be specifically highlighted here.

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

# Appendix

## A. Extended Related Work

### A.1. Curriculum Learning in RL

Automated curriculum design has a long history predating modern LLMs, beginning with classical curriculum learning (Bengio et al., 2009; Graves et al., 2017). These methods assume access to a labeled training set and focus on *reordering* or *selecting* existing data rather than generating new tasks. In the context of RL, curriculum learning helps agents acquire complex behaviors by first mastering simpler tasks (Narvekar et al., 2020; Parashar et al., 2026), or those with high future learning potential (Jiang et al., 2020). Contemporary LLM post-training inherits this paradigm: curriculum is applied over curated prompts or problem categories, using proxy signals such as gradient norms or advantage estimates to guide selection. Examples include synthetic or self-training curricula like Kimi (Team et al., 2025), FastCuRL (Dennis et al., 2020), and LightR1 (Wen et al., 2025), as well as online difficulty-filtering strategies such as Dapo (Yu et al., 2025), Online Difficulty Filtering (Bae et al., 2026), and SEC (Chen et al., 2025b), which discretize problems into difficulty buckets and score

categories by gradient-derived proxies. While these approaches improve learning efficiency in-distribution or OOD, they presuppose that difficulty can be meaningfully partitioned *a priori* and provide only indirect rewards for student progress. Adaptive Data Optimization (ADO) (Jiang et al., 2025) leverages per-domain scaling laws to estimate the learning potential of various data sources online (Jiang et al., 2025). By contrast, our goal is not to arrange data but to elicit learning on a fixed, verifiable hard dataset where standard GRPO fails.

Another line of work explores unsupervised curriculum generation and environment design, moving beyond static datasets by generating a distribution of novel environments suitable for the agent's capabilities (Dennis et al., 2020; Racaniere et al., 2020; Jiang et al., 2020; 2021). Notably, these works find that unconstrained objectives such as minimax adversarial objectives lead to degenerate curricula and reward hacking, and instead optimize regret or estimates of learning potential. We diverge by grounding rewards in student progress on a fixed target set.

### A.2. Self-Play and Teacher-Student Setups

Self-play offers a complementary lens on autonomous capability growth, classically exemplified by game-playing agents trained without external data, such as AlphaZero (Silver et al., 2018). Our approach is inspired by a line of research demonstrating that *asymmetric* self-play can induce powerful automatic curricula. In early work, Sukhbaatar et al. (2018) introduced the canonical Alice–Bob framework in which one agent (Alice) proposes tasks while another (Bob) attempts to solve them, yielding a natural progression of "just-hard-enough" challenges that drive learning. This idea was later extended to complex embodied domains in robotics, where asymmetric self-play enabled automatic discovery of diverse manipulation goals without manual task specification (OpenAI et al., 2021). Applying these ideas from robotics and control to large language models introduces fundamentally different challenges: LLMs operate over a discrete, symbolic problem space with no environment simulator to evaluate intermediate progress; a teacher must generate *entire tasks*, often requiring multi-step reasoning. Moreover, rewards in language domains are extremely sparse and brittle—for mathematical problems, correctness is essentially binary and offers no gradient toward partial solutions. Modern LLM self-play methods thus differ in mechanism: SPIN (Chen et al., 2024), Triplet self-play (Wang et al., 2025), and ReST$^{EM}$ (Singh et al., 2024) optimize for self-consistency or solution quality. These methods generate responses and still presuppose the existence of well-formed input prompts or curated high-quality questions. Recent systems like AlphaProof (Hubert et al., 2025) attempt to mitigate this sparsity at test-time by using an LLM to generate a "natural curriculum" of auxiliary theorem variations for additional training (Hubert et al., 2025). In the context of RLHF, eva (Ye et al., 2024) casts RLHF as an asymmetric creator–solver game in which a creator evolves prompts to expose alignment weaknesses and a solver adapts to reward-model feedback.

A series of near-contemporary works leverages pre-trained LLMs themselves as an untapped resource for question generation. Such "fully data-free" co-evolving systems—including Absolute Zero (Zhao et al., 2025a), R-Zero (Huang et al., 2026), Language Self-Play (LSP) (Kuba et al., 2025), SeRL (Fang et al., 2025) and Self-Questioning Language Models (SQLM) (Chen et al., 2025a)—jointly evolve task creators and solvers via intrinsic or proxy rewards such as majority vote, learnability, reward-model preferences, or gradient magnitudes. Because these methods optimize intrinsic or proxy objectives, they risk drifting to degenerate or unlearnable tasks, are sensitive to reward hacking where models learn to maximize training (pseudo-)reward, and lack guarantees of progress (see an analysis of AbsoluteZero in Chae et al. (2025)). This connects directly to a line of works investigating the broader question of whether self-training — the process where a model learns from its own judgments — can be sustained within RL, and how far self-improvement can be driven by intrinsic or self-generated rewards. Prolonged RL with self-rewards often results in sudden and complete performance collapse (Shafayat et al., 2025; Chae et al., 2025), when rewards vanish or when generator and solver objectives misalign, especially in discrete, symbolic domains with essentially binary correctness signals. Earlier findings in unsupervised curriculum generation likewise note that proxy rewards (such as minimax objectives) can be fragile (Dennis et al., 2020). These observations motivate our design: we learn a teacher *policy* via meta-RL that generates verifiable math questions directly optimized for student learning progress, grounding the curriculum in a concrete failure regime instead of internal proxy of difficulty.

### A.3. Intrinsic Rewards versus Bilevel Optimization

The use of intrinsic rewards is rooted in past studies across robotics, simulation, and task-agnostic settings for curricula generation and exploration. Such rewards maximize objectives such as estimated learning progress, regularity, and prediction error (Schmidhuber, 1991; Pathak et al., 2017; Colas et al., 2019; Blaes et al., 2019; Colas et al., 2022; Sancaktar et al., 2023). To our knowledge, essentially all recent "fully data-free" self-play approaches use intrinsic or proxy rewards to train the teacher/proposer, without anchoring to "real" student performance (with the exception of the self-adaptation work by Zweiger et al. (2025) which uses ReST$^{EM}$/SFT for outer/inner loop). Examples of intrinsic rewards include model confidence

as proposed in Inuitor (Zhao et al., 2026), prediction entropy in DP (Askari-Hemmat et al., 2025), RENT (Prabhudesai et al., 2025), or the majority answer as in TTRL (Zuo et al., 2025) or Shafayat et al. (2025), as well as in SQLM (Chen et al., 2025a).

Of course, the use of proxy rewards is often not merely a design preference but a pragmatic simplification, especially in teacher-student self-play setups: it avoids facing an explicit inner-loop–outer-loop bilevel optimization problem - an appealing but challenging objective where the output of one optimization (in this instance the optimization of the student trained with RLVF on the teacher's question-answer pairs) is fed into another optimization loop (the performance improvement of the student on the hard dataset). Such bilevel optimization objectives have strong historical precedence in meta-learning, in popular methods such as MaML (Finn et al., 2017) and Reptile (Nichol et al., 2018), which explicitly train through an inner-loop–outer-loop structure to obtain efficient few-shot learners, following earlier research like RL2 (Duan et al., 2016), and works that meta-learn hyperparameters of neural nets via full backpropagation through the training loop (Maclaurin et al., 2015).

A similar bilevel formulation, which served as inspiration for our work, also appears in dataset distillation (Wang et al., 2018), where an outer loop optimizes a generally small dataset that allows an inner training loop to achieve good target performance. Here, both proxy-based (e.g., NTK approximation (Nguyen et al., 2021) or feature-matching (Zhou et al., 2022)) and end-to-end bilevel formulations have been explored (Wang et al., 2018; Deng & Russakovsky, 2022; Feng et al., 2024). In general, such approaches become intractable, as the inner loop involves a multi-step computation with a large number of steps, which requires backpropagation through time (BPTT), or "backpropagation through gradient descent", unrolling the inner loop and taking meta-gradients. Our approach, however, avoids the need to unroll the inner loop thanks to the use of RLOO in the outer loop, using the reward (the performance improvement of the student) to reinforce question-answer sets. This is the first instance of a "double meta-RL loop" we are aware of in the context of self-play for LLMs.

# B. Method and Experiment Details

## B.1. Prompts

**Teacher Prompt.** At every outer-loop step, the teacher is given the same prompt. The prompt guides the model towards producing valid math problems using sample subjects/domains and provides explicit instruction regarding the expected format. We avoid seeding the teacher with sample math questions to preserve the data-free setup; the model only sees the black-box reward signal of student performance. We also observe in initial experiments that, when given seed questions, the teacher often collapses to copying them.

```
Teacher Prompt

You are generating a new math problem for a math assistant.

Allowed topics: Algebra, Counting and Probability, Geometry, Intermediate Algebra,
    Number Theory, Prealgebra, or Precalculus.

Output rules (follow EXACTLY):
    - Provide the final formatted problem in this structure: <question>[full math
    question]<question><answer>\\boxed{[answer]}</answer>
    - Any explanations, steps, or reasoning about the problem goes OUTSIDE the
    <question> and <answer> tags.

    Constraints:
        - The problem must be original, challenging, and require at least 2--3
    steps of reasoning.
        - Output exactly ONE problem. You MUST follow the specified format EXACTLY.
Begin now:
```

**Student Prompt.** The same prompt is used for fail@128 filtering, training the student in the inner-loop, and training the student in evaluation.

```
A conversation between User and Assistant. The user asks a question, and the
    Assistant solves it. The assistant first shows the complete reasoning process
    step by step, then provides the final answer in \\boxed{}. The assistant must
    always follow the format: 'User: [question] Assistant: [detailed reasoning] The
    final answer is: \\boxed{[answer]}.
User: <QUESTION> Assistant: "
```

### B.2. Parsing Teacher Outputs

To parse the teacher rollouts into question-answer pairs, we require teacher responses to follow the prompt-specified format. We filter out generations that do not follow this format, and resample until we have $g \cdot n$ correctly-formatted problems. We filter for the following:

- Contains opening and closing question/answer tags.

- Contains the "boxed" notation (denoting an answer).

- Contents of the boxed answer are parsable by a symbolic math verifier.

Theoretically, rejection sampling does not affect the RLOO gradient update (Proposition 1); empirically, we find that this performs better than using teacher-format rewards or sequential question/answer sampling.

**Proposition 1** (RLOO update with rejection sampling). *Let $\pi_0(z)$ be a proposal distribution over some random variable $z$. Let $S$ be a set of "accepted" values of $z$, and assume $\pi_0(S) > 0$. Let*

$$\pi(z) = \pi_0(z)1_{z \in S}/\pi_0(S) \tag{2}$$

*be the distribution on $z$ obtained by rejection sampling, namely, sampling $z$ from $\pi_0$ until $z \in S$.*

*Let $R(z)$ be some reward function on $z$. Then the RLOO update on $\pi$ can be computed from gradient of $\pi_0$ only. Namely, for any $g$-tuple $z_1, \ldots, z_g$ sampled from $\pi$, one has*

$$\sum_{i=1}^{g} A(z_i)\nabla \ln \pi(z_i) = \sum_{i=1}^{g} A(z_i)\nabla \ln \pi_0(z_i) \tag{3}$$

*where*

$$A(z_i) = R(z_i) - \frac{1}{g-1}\sum_{j \neq i} R(z_j) \tag{4}$$

*is the RLOO advantage function, and where the gradients are with respect to the parameters of $\pi$.*

This is not true for simple Reinforce: it relies on the fact that RLOO advantages $A(z_i)$ sum to 0 over $i$.

*Proof.* For any $z$ sampled from $\pi$, one has $z \in S$ with probability 1. For $z \in S$, one has $\ln \pi(z) = \ln \pi_0(z) - \ln \pi_0(S)$. Therefore,

$$\sum_{i=1}^{g} A(z_i)\nabla \ln \pi(z_i) = \sum_{i=1}^{g} A(z_i)\left(\nabla \ln \pi_0(z_i) - \nabla \ln \pi_0(S)\right) \tag{5}$$

$$= \sum_{i=1}^{g} A(z_i)\nabla \ln \pi_0(z_i) - \left(\sum_{i=1}^{g} A(z_i)\right)\nabla \ln \pi_0(S) \tag{6}$$

$$= \sum_{i=1}^{g} A(z_i)\nabla \ln \pi_0(z_i) \tag{7}$$

since the sum of advantages in RLOO satisfies $\sum_i A(z_i) = 0$. $\qquad\square$

## B.3. Reward and Promotion Details

Algorithm 1 details our full algorithm. Note that MA refers to the rolling moving average.

---

**Algorithm 1** SOAR: Teacher-Student meta-RL Training

---

**Input:** Initial teacher $\pi_\phi^T$, initial student $\pi_\theta^S$, threshold $\tau$, group size $g$, dataset size $n$, repeats $r$, MA window size $m$
Initialize timestep $t \leftarrow 0$, MA reward $\bar{R}_0 \leftarrow 0$, $\mathcal{D}_{\text{best}} \leftarrow \emptyset$
**while** $t < T$ **do**
    `// 1.  Teacher generation`
    Sample $g \cdot n$ QA pairs: $\{(q_i, a_i)\}_{i=1}^{g \cdot n} \sim \pi_\phi^T$
    Partition into $g$ datasets: $\mathcal{X}_k = \{(q_j, a_j)\}_{j=n(k-1)+1}^{nk}$ for $k = 1, \ldots, g$
    Sample reward questions $\mathcal{Q}_R = \{(q_j, a_j)\}_{j=1}^M \sim \mathcal{D}_{\text{train}}$
    `// 2.  Inner Loop`
    **for** $k = 1$ **to** $g$ **do**
        **for** $j = 1$ **to** $r$ **do**
            $\theta'_{k,j} \leftarrow$ RLOO-UPDATE$(\theta, \mathcal{X}_k)$                      ▷ **Student RL**
            $R_{k,j} \leftarrow$ ACC$(\theta'_{k,j}, \mathcal{Q}_R) -$ ACC$(\theta, \mathcal{Q}_R)$
        **end for**
        $R_k \leftarrow \frac{1}{r} \sum_{j=1}^r R_{k,j}$
    **end for**
    `// 3.  Check for student promotion.`
    Set $\hat{R}_t \leftarrow \frac{1}{g} \sum_{k=1}^g R_k$
    Update $\bar{R}_t \leftarrow$ MA$(\hat{R}_{t-m}, \ldots, \hat{R}_t)$
    **if** $\bar{R}_t > \tau$ **then**
        $k^* \leftarrow \arg\max_k R_k$
        Find $j^*$ such that $R_{k^*,j^*}$ is the median reward in $\{R_{k^*,j}\}_{j=1}^r$
        $\theta \leftarrow \theta'_{k^*,j^*}$                                 ▷ **Student Promotion**
        $\mathcal{D}_{\text{best}} \leftarrow \mathcal{D}_{\text{best}} \cup \mathcal{X}_{k^*}$
    **end if**
    `// 4.  Teacher Policy Update (Outer-loop)`
    $\phi \leftarrow$ RLOO-UPDATE$(\phi, \{(\mathcal{X}_k, R_k)\}_{k=1}^g)$             ▷ **Teacher RL**
    $t \leftarrow t + 1$
**end while**
**return** $\mathcal{D}_{\text{best}}, \pi_\theta^S$

---

**Stabilizing teacher rewards.** Training inner-loop students with RL can potentially lead to noisy trajectories, and thus noisy teacher rewards. To stabilize the teacher rewards, for each sampled dataset $\mathcal{X}_k$ we execute $r$ parallel student trainings and evaluations, and average their rewards to obtain the final reward: $R_k = \frac{1}{r} \sum_{j=1}^r R_{k,j}$. In practice, we use $r = 4$.

**Promotion mechanism.** At each outer-loop timestep we train $r$ students on each dataset $\mathcal{X}_k$, and "promote" the student baseline when the rolling moving average of aggregated teacher rewards within a fixed window-size exceeds a fixed threshold $\tau$. We choose which trained student to promote by selecting the dataset $\mathcal{X}_k$ with the highest reward $R(\mathcal{X}_k)$ and then selecting the student with the median reward amongst those trained on $\mathcal{X}_k$.

**Computing student rewards.** For inner-loop and evaluation RL on the student, we use the *Math-Verify* package to compare the student-generated and ground-truth answers (Kydlíček, 2025). We assign a reward following standard formulations for RLVR with math:

$$R(y, a) = \begin{cases} 120.0 & \text{if has\_boxed}(y) \wedge \text{verify}(y, a) \\ 20.0 & \text{if has\_boxed}(y) \wedge \neg\text{verify}(\ldots) \wedge a \in y_{ans} \\ 10.0 & \text{if has\_boxed}(y) \wedge \neg\text{verify}(\ldots) \wedge a \notin y_{ans} \\ 0.0 & \text{otherwise} \end{cases}$$

## B.4. Learnability Reward.

To ablate the effects of our grounded reward versus intrinsic rewards, we train teacher models using the well-studied learnability reward (Zhao et al., 2025a; Sukhbaatar et al., 2018). We use the same candidate-generation and dataset-partitioning procedure as SOAR. For each candidate dataset $\mathcal{X}_k = \{(q_i, a_i)\}_{i=1}^n$, we sample 32 completions from the student

for each $q_i$ and compute the average success rate $\bar{s}_i$. The per-question reward is then computed as

$$r_i = \begin{cases} 0, & \text{if } \bar{s}_i = 0 \\ 1 - \bar{s}_i, & \text{otherwise.} \end{cases} \tag{8}$$

We then compute the dataset-level reward as $R_k = \frac{1}{n} \sum_{i=1}^{n} r_i$. For consistency with SOAR, every rollout in $\mathcal{X}_k$ receives the averaged dataset-level reward. We train learnability teachers for 200 steps, and observe convergence of rewards.

### B.5. Datasets

**Fail@128 Filtering.** For each problem in the pool of candidates, we sample 128 solutions with Llama-3.2-3B-Instruct using the student prompt in Appendix B.1, a token budget of 1024 tokens, and temperature 1.0. We keep problems that obtained a 0/128 success rate.

**OlympiadBench.** For OlympiadBench, we source our fail@128 questions from the subset that is in English, text-only, and automatically verifiable (674 total questions). Since OlympiadBench was originally designed as a test set, we construct a random train/test split.

**HARP.** We source our fail@128 problems from the full HARP dataset. Since HARP was originally designed as a test set, we construct a random train/test split.

**MATH.** In preliminary experiments, we observed a large gap between the zero-shot accuracy of Llama-3.2-3B-Instruct on the official MATH training vs. test splits (60% vs. 37%), suggesting that the model may have partial exposure to the MATH training questions. To minimize confounding effects from such memorization, we draw our initial pool of hard problems from the 5000-problem official MATH test split. We then apply the fail@128 filter and construct our own internal train/test split from this filtered subset. All synthetic data generation and student-teacher training uses only the internal training split, and final results are reported exclusively on the held-out internal test split.

**Dataset sizes.** In Table 2 we report the original size of each problem pool, and the sizes of our train/test splits.

*Table 2.* **Dataset sizes pre- and post- fail@128 filtering.**

| Dataset | Initial problem pool | fail@128 train set | fail@128 test set |
|---|---|---|---|
| MATH | 5000 | 359 | 360 |
| HARP | 4768 | 714 | 714 |
| Olympiad Bench | 674 | 158 | 158 |

### B.6. Evaluation

**Mixed synthetic-real training.** We primarily evaluate generated questions by training a fresh student model on a combination of the synthetic questions, and the real fail@128 train set. We explore two mixing strategies:

- **Curriculum training.** We first train the student on synthetic questions for a fixed number of training steps (64), and then switch to training on real fail@128 training questions, aiming to mirror the trajectory of training a promoted student. Here, the synthetic questions act as a "warm-start", enabling the student to obtain gradient signal on the harder problems. The synthetic training window was chosen as a representative budget based on preliminary experiments.

- **Mixed training.** We train on a mixture of synthetic and real questions throughout.

To avoid biasing results, we select between curriculum/mixed training using our baseline methods.

On MATH, while both exhibit similar training dynamics, we found that our *Base-T* baseline performed better with curriculum and thus adopt it for all MATH experiments (Figure 7). On OlympiadBench and HARP we observed that mixed training yields significantly more stable learning dynamics, even when adding real instead of synthetic data. Figure 8 compares mixed/curriculum training on HARP and OlympiadBench fail@128 with 128 real MATH problems. Curriculum training

exhibits an early performance spike, followed by a significant and sudden performance decline early in training. Thus for HARP and OlympiadBench we use mixed training in our evaluations.

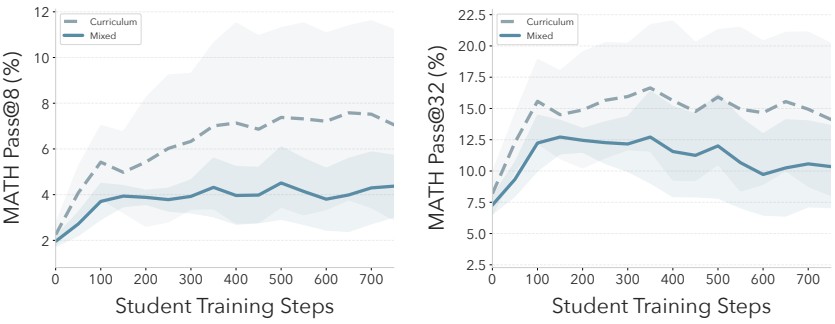

*Figure 7.* **Mixed v. Curriculum training on MATH.** We compare training the base student on MATH fail@128 + 128 questions sampled from *Base-T*. Curriculum performs better across different inference budgets.

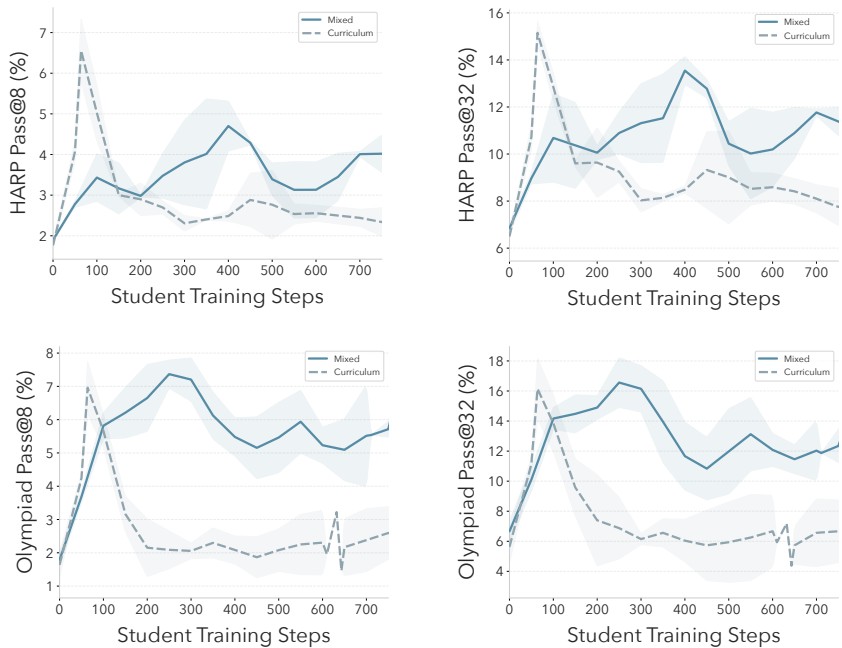

*Figure 8.* **Mixed v. Curriculum training on HARP/OlympiadBench.** We compare training the base student on real fail@128 + 128 random MATH questions, for HARP and OlympiadBench. Mixed training exhibits significantly more stable training dynamics across inference budgets (Pass@8 and Pass@32) and converges to higher final performance points. For both datasets, curriculum training exhibits more instability with a large early performance spike and then crash.

**Teacher sampling.** At evaluation time, we sample problems from the trained teacher using the same prompt and format-filtering as in training.

**PQ/PS Evaluation.** We evaluate PQ using mixed synthetic/real training, described above. We evaluate PS by simply running inference on the fail@128 test set, to evaluate how much the student baseline advanced during SOAR training.

**Student checkpoint selection.** For evaluations involving fresh student models, we train for a maximum of 1500 steps (typically observing convergence well before this point). For MATH and HARP experiments where we report performance at a fixed point, we select the student checkpoint to evaluate at using the *slope of the smoothed training reward curve*, similarly to classic RL early stopping heuristics (Mahsereci et al., 2017). In particular, we smooth the average training reward curve (centered-moving-average, 25 steps) and compute the discrete slopes, normalized by the range of observed rewards. The early stopping step is defined as the earliest point where the normalized slope falls below 15% of the maximum observed slope. We selected a 15% threshold to identify the beginning of the reward plateau; empirically, varying between

10% and 20% have negligible effects on the selected point. Test performance is averaged over a 200 step window following the selected step, to account for variance. We also show the full training curves in Section C.1.

We choose this heuristic to account for differing convergence rates between methods on MATH and HARP, and our small dataset sizes. In initial experiments we found separate validation sets, and cross-validation with the train set, to be extremely noisy. On OlympiadBench we observe similar convergence across all methods, and report at a fixed point of 50 steps.

### B.7. Hyperparameters

In Table 3 we detail our training and evaluation hyperparameters.

**Outer-loop training.** We performed the following sweeps in preliminary experiments, and tuned using student performance on the full train set. Once selected, the same hyperparameters are used across all training runs and datasets. See Appendix D.2 for ablations on sensitivity to threshold $\tau$ and dataset size $n$.

- LR: {1e-6, 5e-6, **1e-5**, 5e-5}
- $n$: {8, 16, 32, **64**}
- $\tau$: {**0.01**, 0.015, 0.02}
- Moving avg window size: {1, **3**}

We train for a maximum of 200 outer steps based on compute constraints. For teacher-sampling experiments we fix the evaluation checkpoint based on the point of decline of teacher rewards observed in initial runs (170 steps for all HARP-trained models, 200 steps for all MATH-trained models).

**Inner-loop training.** We find that from the base student, 10 steps is sufficient to induce movement in student performance. As the student baseline is updated, it is helpful to train slightly longer (we use +5 steps). We use greedy decoding for evaluating on $\mathcal{Q}_R$ to reduce noise in the student reward.

**Evaluation.** We use standard hyperparameters to train the student from scratch on combined real/synthetic data (Table 3c). For PQ with curriculum evaluation we use zero learning rate warmup to match the inner-loop environment.

### B.8. Seeds

To ensure statistical significance and account for both teacher-training and student-training variation, we employ a nested seeding strategy.

**Teacher training.**

- For our main SOAR experiments, we train four independent teachers each on MATH and HARP to cover a range of teacher training outcomes.
- For teacher objective ablations (*Intrinsic-T* and *Grounded-T (no promotion)*) we trained three independent teachers each.

**Evaluation (student training).**

- The *Hard-Only* baseline is evaluated over $\geq 6$ student seeds.
- For PQ datasets (>2 promotions), we train at least three students per PQ dataset, totaling $\geq 6$ seeds (2 PQ datasets $\times$ 3 students) per reported metric.
- For PS students, we compute pass@$k$ metrics using inference over three seeds.
- For teacher-sampling experiments (*i.e.,* sampling data from trained teachers and then training a fresh student) we train 2-3 independent students per teacher seed, resulting in $\geq 8$ seeds per reported metric.

For all metrics we report the aggregated mean and standard deviation over student seeds.

*Table 3.* **Hyperparameters for SOAR training and evaluation.**

| Hyperparameter | Teacher | Student |
|---|---|---|
| Optimizer | AdamW | |
| KL coefficient | 0.001 | |
| LR schedule | Cosine decay | |
| Learning rate | 1e-5 | |
| Temperature | 1.0 | |
| LR warmup steps | 20 | 0/20 |
| Batch size | 2 | 8 |
| Group size | 4 | 32 |
| Max generated tokens | 512 | 1024 |
| *meta-RL specific (teacher only)* | | |
| Promotion threshold ($\tau$) | 0.01 | — |
| Moving avg window | 3 | — |
| Dataset size ($n$) | 64 | — |
| Student repeats ($r$) | 4 | — |
| *Evaluation specific (student only)* | | |
| Max training steps | — | 1500 |
| Synthetic warmup steps (curriculum training) | — | 64 |

## B.9. Computational Resources

Each SOAR *training* run was executed on 4 nodes (each $8\times$ NVIDIA H200 GPUs or $8\times$ NVIDIA H100 GPUs) for $\approx$ 48-60 hours. Each RLOO *evaluation* run (training a fresh student) was executed for $\approx$ 12 hours on 1 H200 node or 1 H100 node.

## C. Expanded Main Experiments

### C.1. Full Student Training Curves

In Figure 9 we show full student training curves for PQ, *Hard-Only*, and the full MATH upper bound for MATH, HARP, and OlympiadBench. In Figures 11-13 we show these training curves for questions sampled from *Grounded-T*, *Base-T*, *Intrinsic-T*, and *Grounded-T (no promotion)*. All curves show the mean and standard deviation over seeds.

### C.2. Full Evaluations on fail@128 MATH, HARP, and OlympiadBench.

In Tables 4-5 we report our full results from evaluating SOAR on MATH and HARP (in-domain datasets). In Table 6 we report full results from evaluating on OlympiadBench, an OOD dataset.

Our PQ datasets have one of $\{128, 192, 256\}$ questions, depending on the number of student promotions for each run. For *Intrinsic-T* we sample 128 questions, consistent with all of our teacher-sampling experiments. For the equal-data comparison between *Intrinsic-T* and *Grounded-T* (sampling from the SOAR-trained teacher), see Section 5.2 and Appendix C.3.

In addition to the methods/baselines shown in Figure 3 we also report the following.

**Inference pass@k with the base model.** Inference with the base model has non-zero pass@$k$ due to stochastic sampling with different seeds than were used for the initial pass@128 = 0 filtering. Comparison with *Hard-Only* results shows that our fail@128 datasets are sufficiently difficult such that direct training yields very little improvement.

**Inference pass@k with the trained teacher.** We perform direct inference on fail@128 problems with final trained teacher policies, denoted as "Inference with *Grounded-T* " in Tables 4-5). We find no improvement over the base model, indicating that improved *teaching ability* is decoupled from improved *solving ability*.

***Hard-Only* with extra compute.** A natural question is whether we can improve direct training on fail@128 train questions simply by increasing compute. One strategy is to train for longer, however our learning curves in Figure 9 show that *Hard-Only* test performance decreases in the latter stages of training. We confirm this by extending *Hard-Only* training with HARP from 1500 to 6500 steps (Figure 10). Another strategy is to sample more from the base model by increasing the RLOO group size. On MATH, we increase the group size $4\times$ (from our default $g = 32$ to $g = 128$), and find that it only

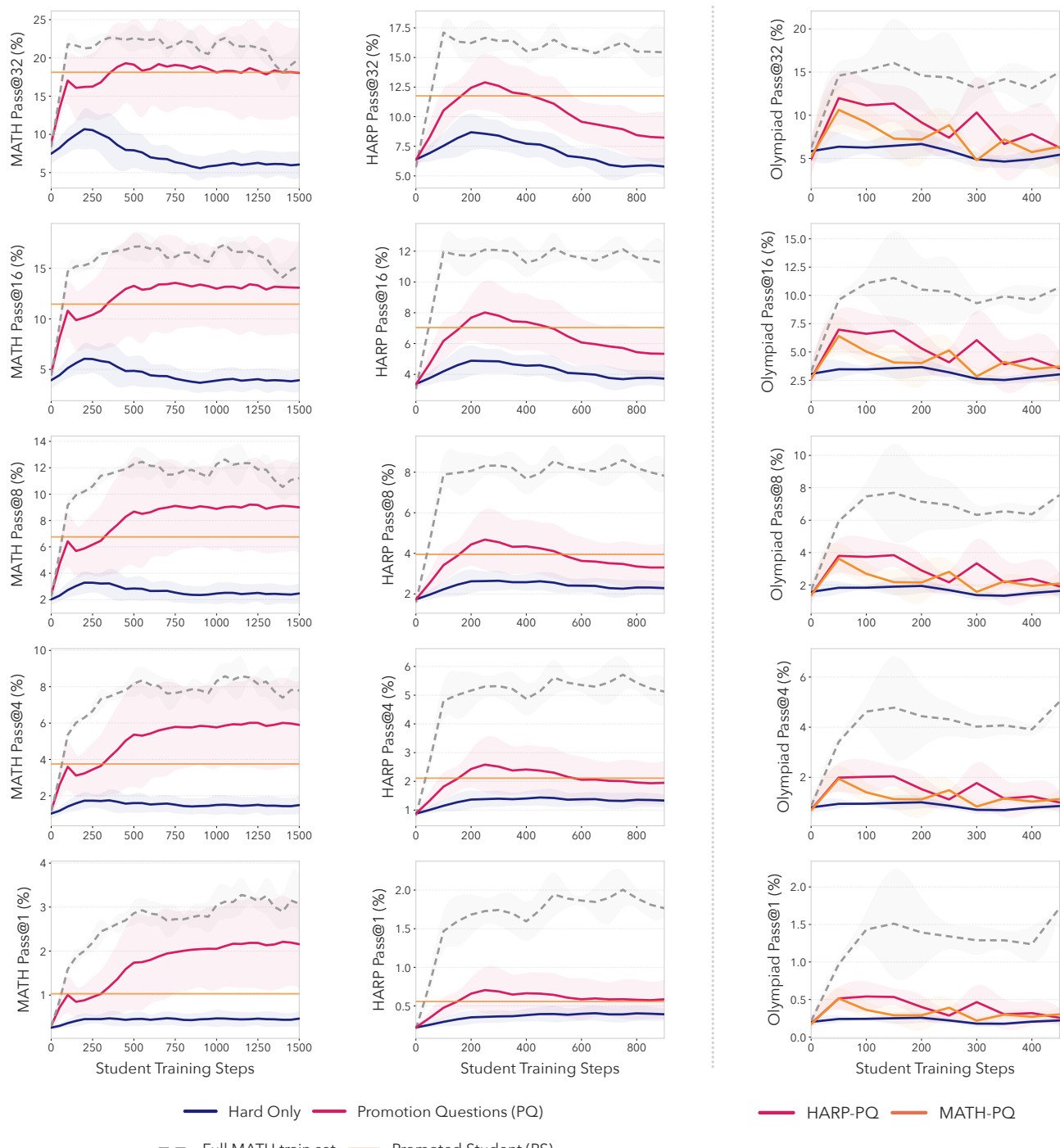

*Figure 9.* **Fail@128 test performance during student training for MATH, HARP, and Olympiad.** Student learning curves for different pass@k when trained on *Hard-Only*, PQ, or the Full MATH dataset (PS inference performance shown as a horizontal line). PQ and PS improve performance on all inference budgets and datasets, with increased effect at higher $k$. On MATH, PQ exhibits performance gains even after the synthetic-training phase (64 steps), showing that synthetic problems make real hard problems more learnable.

yields marginal improvements over *Hard-Only* (*e.g.*, +2.8% pass@32) and does not recover the improvements of PQ.

**SeRL.** We evaluate SeRL ([Fang et al.](#), 2025) as a contemporary self-play baseline. SeRL self-evolves a curriculum using an initial seed dataset with the goal of outperforming direct RLVR on that dataset. SeRL generates question-answer pairs

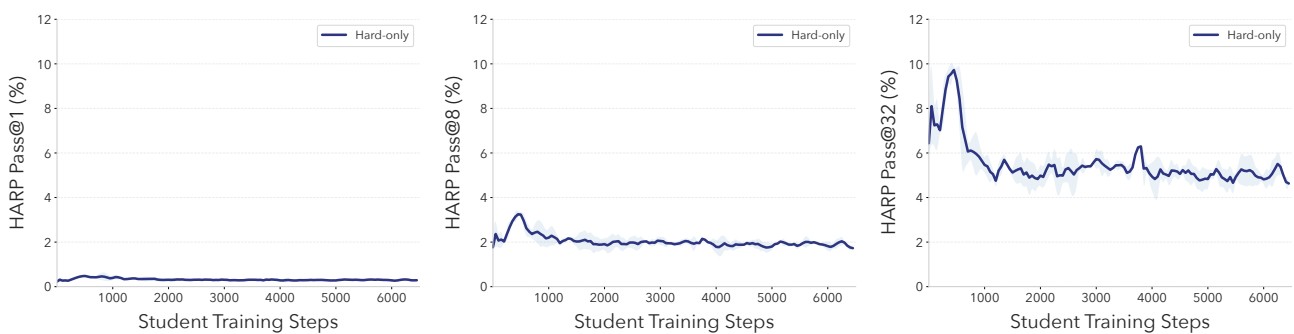

*Figure 10.* **Extended *Hard-Only* training on HARP.** We extend *Hard-Only* training on HARP from 1500 steps to 6500 steps, and find that the extra compute does not improve performance. Shading shows $\pm$ 1 SD over 3 seeds.

*Table 4.* **MATH pass@k (%) test accuracy on fail@128.** Mean and SD over seeds are reported at the timestep determined by training reward convergence (see Appendix B.6) with full curves in Figure 9. PQ and PS consistently outperform inference-only, *Hard-Only*, and intrinsic baselines across all inference budgets, and recover the majority of performance gain from training with real curated problems. We boldface the best among "data-free" methods (*i.e., only* $\mathcal{D}_{train}$ *available*). The bottom three rows serve as upper bounds from using curated, expert-annotated data. PQ datasets contain one of $\{128, 192, 256\}$ questions.

| | k | | | | |
|---|---|---|---|---|---|
| **Method** | **1** | **4** | **8** | **16** | **32** |
| Base Model Inference | $0.3 \pm 0.1$ | $1.0 \pm 0.2$ | $2.0 \pm 0.4$ | $3.9 \pm 0.8$ | $7.5 \pm 1.3$ |
| *Hard-Only* | $0.5 \pm 0.1$ | $1.7 \pm 0.4$ | $3.2 \pm 0.8$ | $5.7 \pm 1.5$ | $9.6 \pm 2.6$ |
| *Hard-Only* ($g = 128$) | $1.4 \pm 1.0$ | $3.9 \pm 2.6$ | $6.1 \pm 3.9$ | $8.9 \pm 5.5$ | $12.4 \pm 7.4$ |
| SeRL (step 50) | $0.5 \pm 0.1$ | $1.9 \pm 0.3$ | $3.7 \pm 0.4$ | $6.9 \pm 0.7$ | $12.2 \pm 0.7$ |
| SeRL (step 100) | $0.5 \pm 0.0$ | $2.1 \pm 0.0$ | $4.0 \pm 0.0$ | $7.3 \pm 0.2$ | $12.4 \pm 1.0$ |
| SOAR-PQ (Ours) | $\mathbf{1.7 \pm 1.0}$ | $\mathbf{5.3 \pm 2.6}$ | $\mathbf{8.5 \pm 3.7}$ | $13.0 \pm 4.8$ | $18.9 \pm 5.3$ |
| SOAR-PS (Ours) | $1.0 \pm 0.2$ | $3.8 \pm 0.6$ | $6.8 \pm 1.1$ | $11.5 \pm 1.6$ | $18.1 \pm 2.4$ |
| *Grounded-T* (Ours) | $1.6 \pm 0.5$ | $5.1 \pm 1.4$ | $8.4 \pm 2.1$ | $\mathbf{13.1 \pm 2.9}$ | $\mathbf{19.1 \pm 3.7}$ |
| *Intrinsic-T* | $1.0 \pm 0.6$ | $3.3 \pm 2.1$ | $5.7 \pm 3.5$ | $9.2 \pm 5.3$ | $14.1 \pm 7.5$ |
| Inference with *Grounded-T* | $0.3 \pm 0.0$ | $1.1 \pm 0.2$ | $2.2 \pm 0.3$ | $4.4 \pm 0.6$ | $8.3 \pm 1.1$ |
| HARP train (128) | $2.4 \pm 1.0$ | $7.2 \pm 2.4$ | $11.3 \pm 3.1$ | $16.5 \pm 3.6$ | $23.0 \pm 3.9$ |
| MATH train (128) | $2.1 \pm 0.0$ | $6.6 \pm 0.1$ | $10.5 \pm 0.3$ | $15.7 \pm 0.5$ | $21.8 \pm 0.9$ |
| MATH train (Full) | $2.7 \pm 0.2$ | $7.6 \pm 0.7$ | $11.5 \pm 1.2$ | $16.4 \pm 1.8$ | $22.0 \pm 2.4$ |

with majority-voting self-rewards, and filters based on learnability and diversity. We train SeRL with MATH and HARP fail@128 respectively as the seed sets, and observe convergence after 100 steps. As `Llama-3.2-3B-Instruct` is a primary model used in (Fang et al., 2025), we adopt their reported hyperparameters. Results are shown in Tables 4 and 5; SeRL improves over *Hard-Only* however fails to match SOAR.

**Sampling curated "oracle questions".** In addition to training with the full MATH train set, we also evaluate sampling 128 questions from the MATH and HARP train sets, which can be considered oracle (curated/expert-annotated) data sources. We choose 128 to match our teacher sampling experiments (Section C.3) and roughly match the amount of PQ data, which varies between 128 and 256 questions.

On MATH, training with these smaller subsets performs similarly to training with the full MATH dataset, suggesting a saturation point. On HARP, these smaller subsets only recover $\approx 50\%$ of the gains from training with the full MATH train set. Notably, PQ and PS both outperform 128 sampled questions from HARP, and match 128 questions from MATH.

*Table 5.* **HARP pass@k (%) test accuracy on fail@128**. Mean and SD over seeds are reported at the timestep determined by training reward convergence (see Appendix B.6) with full curves in Figure 9. PQ and PS consistently outperform inference-only, *Hard-Only*, and intrinsic baselines across all inference budgets. Notably, SOAR questions perform better on HARP than similar numbers of questions from the MATH/HARP datasets (which serve as a curated, expert-annotated data source).

| Method | k | | | | |
|---|---|---|---|---|---|
| | **1** | **4** | **8** | **16** | **32** |
| Base Model Inference | $0.2 \pm 0.0$ | $0.9 \pm 0.0$ | $1.7 \pm 0.0$ | $3.4 \pm 0.0$ | $6.4 \pm 0.0$ |
| *Hard-Only* | $0.4 \pm 0.1$ | $1.4 \pm 0.2$ | $2.6 \pm 0.4$ | $4.7 \pm 0.6$ | $8.2 \pm 1.0$ |
| SeRL (step 50) | $0.4 \pm 0.0$ | $1.4 \pm 0.1$ | $2.7 \pm 0.3$ | $5.3 \pm 0.5$ | $9.7 \pm 0.9$ |
| SeRL (step 100) | $0.5 \pm 0.0$ | $1.9 \pm 0.1$ | $3.6 \pm 0.2$ | $6.5 \pm 0.3$ | $10.9 \pm 0.4$ |
| SOAR-PQ (Ours) | $\mathbf{0.7 \pm 0.3}$ | $\mathbf{2.5 \pm 0.8}$ | $\mathbf{4.5 \pm 1.3}$ | $\mathbf{7.7 \pm 1.7}$ | $\mathbf{12.3 \pm 2.0}$ |
| SOAR-PS (Ours) | $0.6 \pm 0.1$ | $2.1 \pm 0.3$ | $3.9 \pm 0.6$ | $7.0 \pm 0.9$ | $11.8 \pm 1.2$ |
| *Grounded-T* (Ours) | $0.5 \pm 0.2$ | $2.0 \pm 0.5$ | $3.8 \pm 0.9$ | $6.7 \pm 1.3$ | $11.2 \pm 1.7$ |
| *Intrinsic-T* | $0.4 \pm 0.1$ | $1.6 \pm 0.5$ | $3.1 \pm 0.8$ | $5.6 \pm 1.4$ | $9.6 \pm 2.1$ |
| Inference with *Grounded-T* | $0.2 \pm 0.0$ | $0.9 \pm 0.1$ | $1.9 \pm 0.2$ | $3.6 \pm 0.4$ | $6.8 \pm 0.7$ |
| HARP train (128) | $0.4 \pm 0.0$ | $1.4 \pm 0.1$ | $2.8 \pm 0.2$ | $5.0 \pm 0.5$ | $8.7 \pm 1.1$ |
| MATH train (128) | $0.6 \pm 0.1$ | $2.1 \pm 0.4$ | $4.0 \pm 0.7$ | $7.1 \pm 0.9$ | $11.9 \pm 0.9$ |
| MATH train (Full) | $1.7 \pm 0.2$ | $5.1 \pm 0.4$ | $8.1 \pm 0.4$ | $11.7 \pm 0.3$ | $16.2 \pm 0.4$ |

### C.3. Sampling from Teacher Models.

While PQ comes from accumulated useful questions over the meta-RL trajectory, here we *sample questions directly from the trained teacher policy*. The similar performance of *Grounded-T* and PQ (Tables 4-5) provide evidence that the pedagogical signals captured in the PQ datasets are learned by the teacher's distribution.

In Figures 11-13 we show full test trajectories on MATH, HARP, and Olympiad for students trained with 128 questions sampled from *Grounded-T*, *Intrinsic-T*, *Base-T*, and *Grounded-T (no promotion)*. *Grounded-T* outperforms all comparisons, particularly at higher inference budgets, and is competitive with PQ. *Grounded-T* also exhibits lower variance and greater stability across student and teacher seeds. *Grounded-T (no promotion)* performs worse than *Grounded-T*, PQ, and PS, validating the importance of the promotion mechanism.

In Figure 14 we also compare student trajectories for each *Grounded-T* and *Intrinsic-T* teacher seed. Consistent with MATH and HARP (Figure 5), students have similar trajectories across independent *Grounded-T* teachers, and high variance across different *Intrinsic-T* teachers, showcasing the instability of intrinsic rewards.

### C.4. Categorizing Correctness of Synthetic Questions.

We categorize synthetic questions into *correctness taxonomies* using `Claude-4.5-Sonnet` as an oracle judge. The prompt given to Claude is shown below. In Table 7 we report taxonomy statistics for PQ datasets, and problems sampled from *Grounded-T*, *Intrinsic-T*, and *Base-T* teachers.

We prompt `Claude-4.5-Sonnet` to categorize problems as follows:

- `Well posed`: If the problem is mathematically complete and solvable.

- `Correct`: If the proposed answer is correct (only if the problem is well posed).

- `Error type`:
  - `None`
  - `Arithmetic error`: Sound logic, but incorrect final calculation.
  - `Logical fallacy`: Does not follow mathematical rules.
  - `Ill-posed/Impossibility`: The question contains a mathematical impossibility.

*Table 6.* **Olympiad pass@k (%) test accuracy on fail@128.** Mean and SD over seeds are reported timestep 50 with full curves in Figure 9. Despite being optimized with reward signals from HARP and MATH, PQ questions and PS inference transfer to improving performance on Olympiad, and match or outperform 128 questions sampled from the HARP train set (a curated/expert-annotated source of problems). PS and PQ transfer better when trained with HARP than with MATH, potentially indicating more shared structure between HARP and Olympiad.

| | **k** | | | | |
|---|---|---|---|---|---|
| **Method** | **1** | **4** | **8** | **16** | **32** |
| Base Model Inference | $0.2 \pm 0.0$ | $0.8 \pm 0.1$ | $1.6 \pm 0.3$ | $3.1 \pm 0.5$ | $5.8 \pm 1.0$ |
| *Hard-Only* | $0.3 \pm 0.1$ | $1.1 \pm 0.3$ | $2.1 \pm 0.6$ | $3.9 \pm 1.3$ | $6.9 \pm 2.7$ |
| SOAR-PQ (MATH) (Ours) | $0.5 \pm 0.1$ | $1.9 \pm 0.5$ | $3.6 \pm 0.9$ | $6.4 \pm 1.6$ | $10.6 \pm 2.7$ |
| SOAR-PQ (HARP) (Ours) | $0.5 \pm 0.1$ | $2.0 \pm 0.5$ | $\mathbf{3.8 \pm 1.0}$ | $\mathbf{7.0 \pm 1.8}$ | $\mathbf{12.0 \pm 3.0}$ |
| SOAR-PS (MATH) (Ours) | $\mathbf{0.6 \pm 0.1}$ | $\mathbf{2.1 \pm 0.5}$ | $3.7 \pm 0.8$ | $6.2 \pm 1.3$ | $9.9 \pm 2.2$ |
| SOAR-PS (HARP) (Ours) | $0.5 \pm 0.1$ | $2.0 \pm 0.4$ | $\mathbf{3.8 \pm 0.7}$ | $6.9 \pm 1.1$ | $11.7 \pm 1.6$ |
| *Grounded-T* (MATH) (Ours) | $0.4 \pm 0.2$ | $1.6 \pm 0.8$ | $2.9 \pm 1.4$ | $5.3 \pm 2.4$ | $9.0 \pm 4.0$ |
| *Grounded-T* (HARP) (Ours) | $0.5 \pm 0.2$ | $1.9 \pm 0.6$ | $3.6 \pm 1.1$ | $6.5 \pm 1.8$ | $11.1 \pm 2.9$ |
| *Intrinsic-T* | $0.4 \pm 0.3$ | $1.7 \pm 1.2$ | $3.1 \pm 2.0$ | $5.5 \pm 3.4$ | $9.1 \pm 5.2$ |
| HARP train (128) | $0.5 \pm 0.1$ | $2.0 \pm 0.2$ | $3.6 \pm 0.4$ | $6.5 \pm 0.8$ | $10.6 \pm 1.7$ |
| MATH train (128) | $1.0 \pm 0.1$ | $3.4 \pm 0.1$ | $5.9 \pm 0.1$ | $9.6 \pm 0.4$ | $14.6 \pm 1.4$ |
| MATH train (Full) | $0.9 \pm 0.0$ | $3.2 \pm 0.1$ | $5.6 \pm 0.3$ | $8.8 \pm 0.7$ | $13.1 \pm 0.9$ |

– `Ambiguous`: The question is missing data, variables, or context necessary for solving it.

Our results show that the well-posedness of a problem matters more than the correctness of the solution. While teacher-training improves the correctness rate, the best-performing datasets (*Grounded-T* and PQ) only contain 32.8% and 36.5% correct solutions respectively, compared to 55.5% for *Intrinsic-T*. This indicates that question diversity is more important for success (see Table 1). Meta-RL mainly reduces question ambiguities (improving well-posedness) while the rate of arithmetic errors remains the same or slightly higher.

### C.5. Do Incorrect Synthetic Questions Help?

We run a controlled experiment to isolate the effects of training on well-posed questions with incorrect answers. We compare training a fresh student on the subset of PQ-HARP questions with correct answers (*Correct-only*; 82 questions), and the subset with well-posed questions (*Well-posed*; 144 questions). Note that *Correct-only* is strictly a subset of *Well-posed*. In both cases, we follow the standard HARP evaluation protocol and train on a mix of synthetic and real fail@128 questions.

Results are shown in in Figure 15. Adding well-posed questions with incorrect answers improves performance, showing that performance gains are not driven solely by correct questions. We further note that across all experiments (*Hard-Only*, *Correct-only*, *Well-posed*) the student format match reaches 100% within the first $\sim 50$ steps; thus, formatting is also not the main driver of performance gains.

You are evaluating generated math problems for their coherence and solvability. Your task is to determine if the given question is well-formulated, and if the given answer is correct.

CRITICAL INSTRUCTION: Do not assume missing information. If the question is nonsensical, lacks a clear problem/question/equation, is syntactically incorrect, is missing necessary information, or is missing variables, you MUST classify it as 'Ambiguous' or 'Ill_Posed'. Do not invent a context to make the answer work.

QUESTION: {question}
PROPOSED_ANSWER: {proposed_answer}

TAXONOMY OF ERRORS:
- 'None': The question is mathematically complete and the answer is correct.
- 'Arithmetic': The logic is sound, but the final calculation is wrong.
- 'Logical_Fallacy': The steps taken do not follow mathematical rules.
- 'Ill_Posed': The question contains a mathematical impossibility.
- 'Ambiguous': The question is missing necessary data, variables, or context (e.g., "Solve the equation" without providing the equation).

TASK:
1. Analyze the QUESTION for completeness. If it's a "fragment" or "nonsense," stop and flag it.
2. Solve the problem ONLY if it is well-defined.
3. Determine:
   - is_well_posed: boolean - Is the question mathematically complete and solvable?
   - is_correct: boolean - Is the proposed answer correct? (Only evaluate if is_well_posed is true)
   - error_type: one of ['None', 'Arithmetic', 'Logical_Fallacy', 'Ill_Posed', 'Ambiguous']
   - verified_answer: string - The correct answer if the question is well-posed, or "N/A" if not well-posed

OUTPUT FORMAT:
First, provide your reasoning in <think> tags.
Then, provide a JSON object with the following exact structure:

```json
{{
    "is_correct": <boolean>,
    "is_well_posed": <boolean>,
    "error_type": "<one of: None, Arithmetic, Logical_Fallacy, Ill_Posed, Ambiguous>",
    "verified_answer": "<string: the correct answer or 'N/A'>"
}}
``

EXAMPLE OUTPUT:
<think>
The question asks to solve 2x + 5 = 13. This is well-posed with all necessary information. Solving: 2x = 8, so x = 4. The proposed answer is 4, which is correct.
</think>

```json
{{
    "is_correct": true,
    "is_well_posed": true,
    "error_type": "None",
    "verified_answer": "4"
}}
```

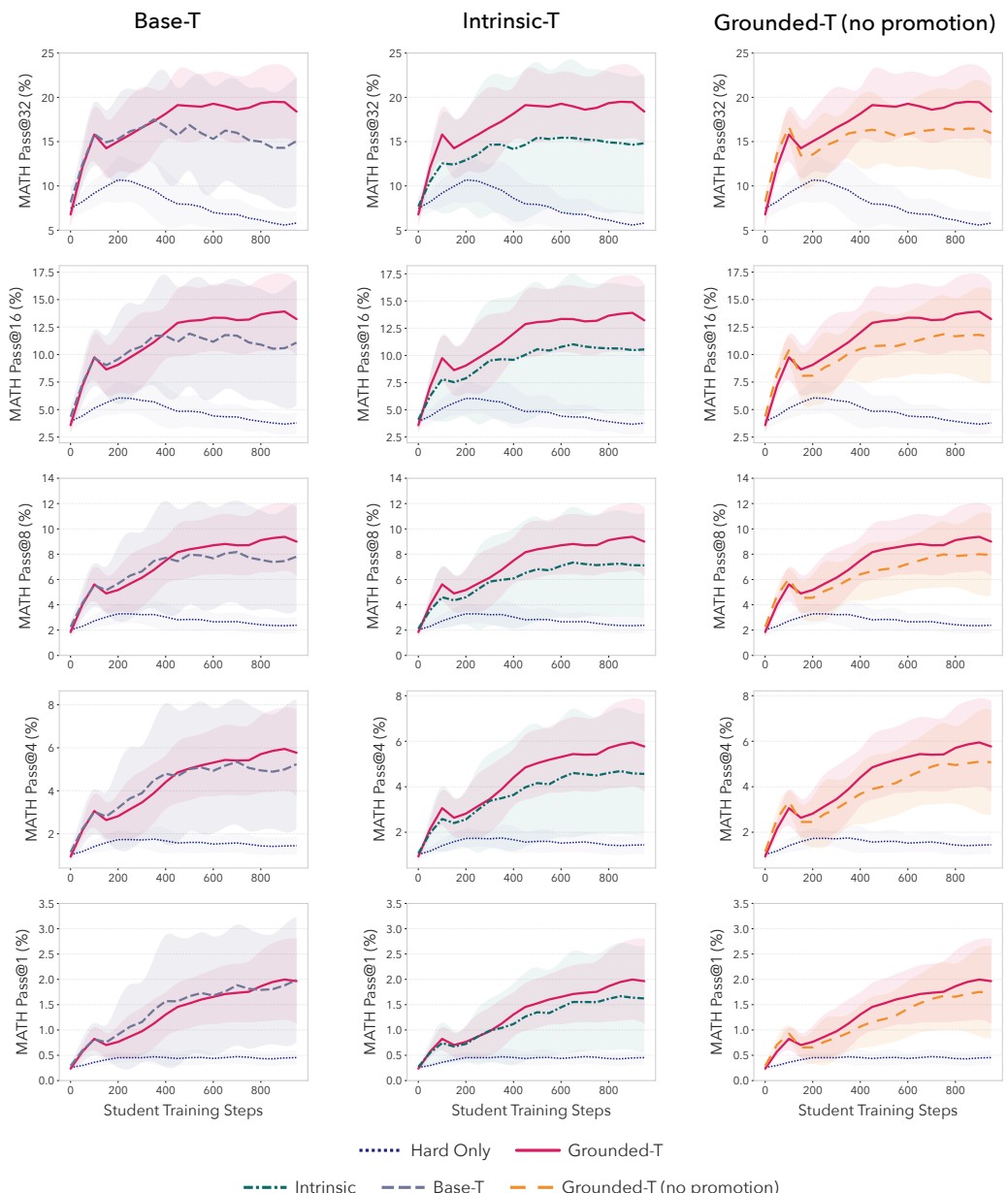

*Figure 11.* **Fail@128 test performance during student training for MATH with different teachers.** Each column compares training a fresh student with 128 questions from *Grounded-T* to 128 questions from a different teacher (*Hard-Only* also included for reference). While all teachers outperform *Hard-Only*, *Grounded-T* performs best, with increasing effects at higher *k*. *Grounded-T* results in less variance across student outcomes, particularly compared to *Base-T* and *Intrinsic-T*. PQ learning curves are in Figure 9.

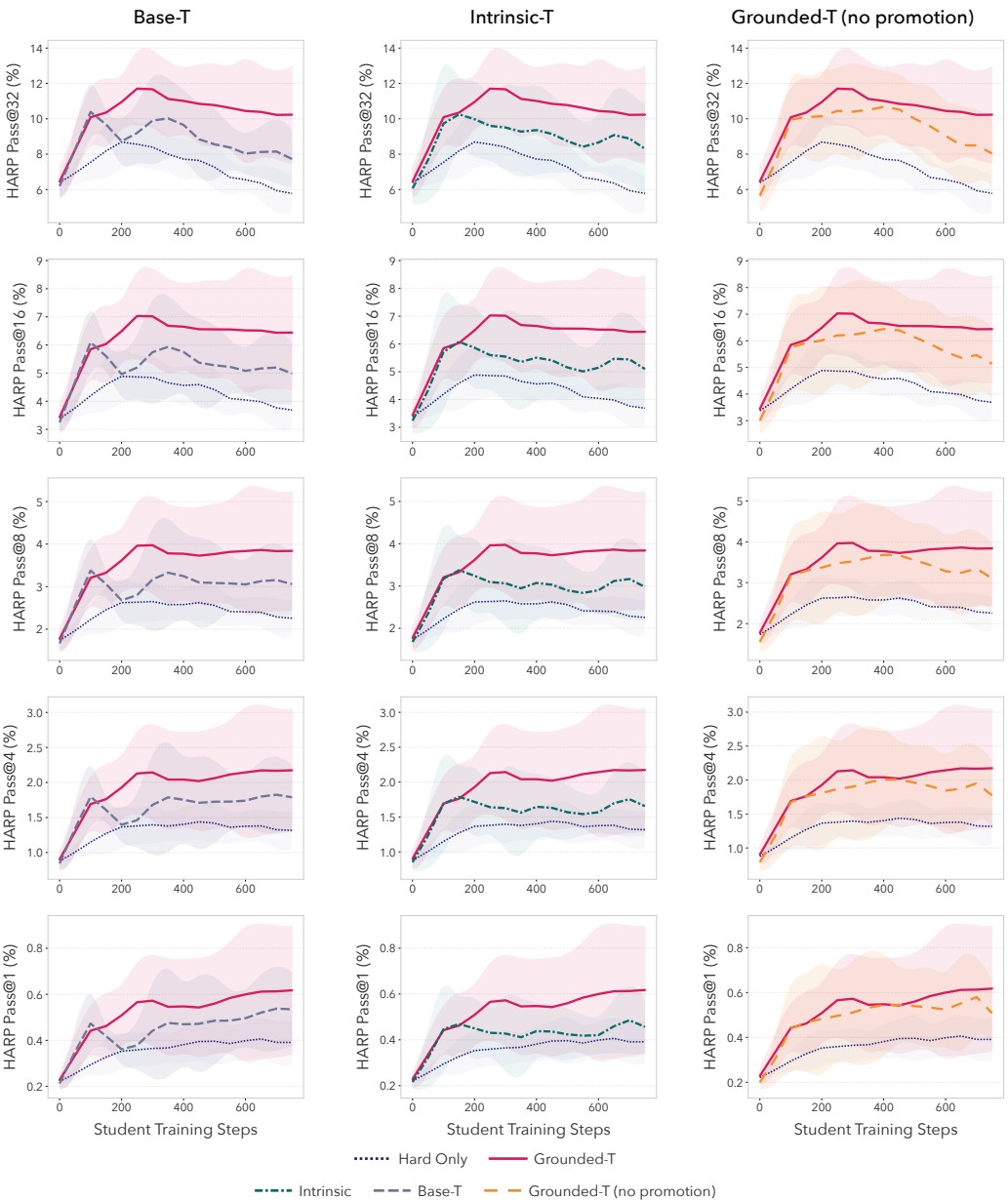

*Figure 12.* **Fail@128 test performance during student training for HARP with different teachers.** Each column compares training a fresh student with 128 questions from *Grounded-T* to 128 questions from a different teacher (*Hard-Only* also included for reference). *Grounded-T* performs best, with increasing effects at higher $k$. Students trained with *Base-T* and *Intrinsic-T* tend to decline more for higher $k$ in the later stages of training, while *Grounded-T* leads to more stable trajectories.

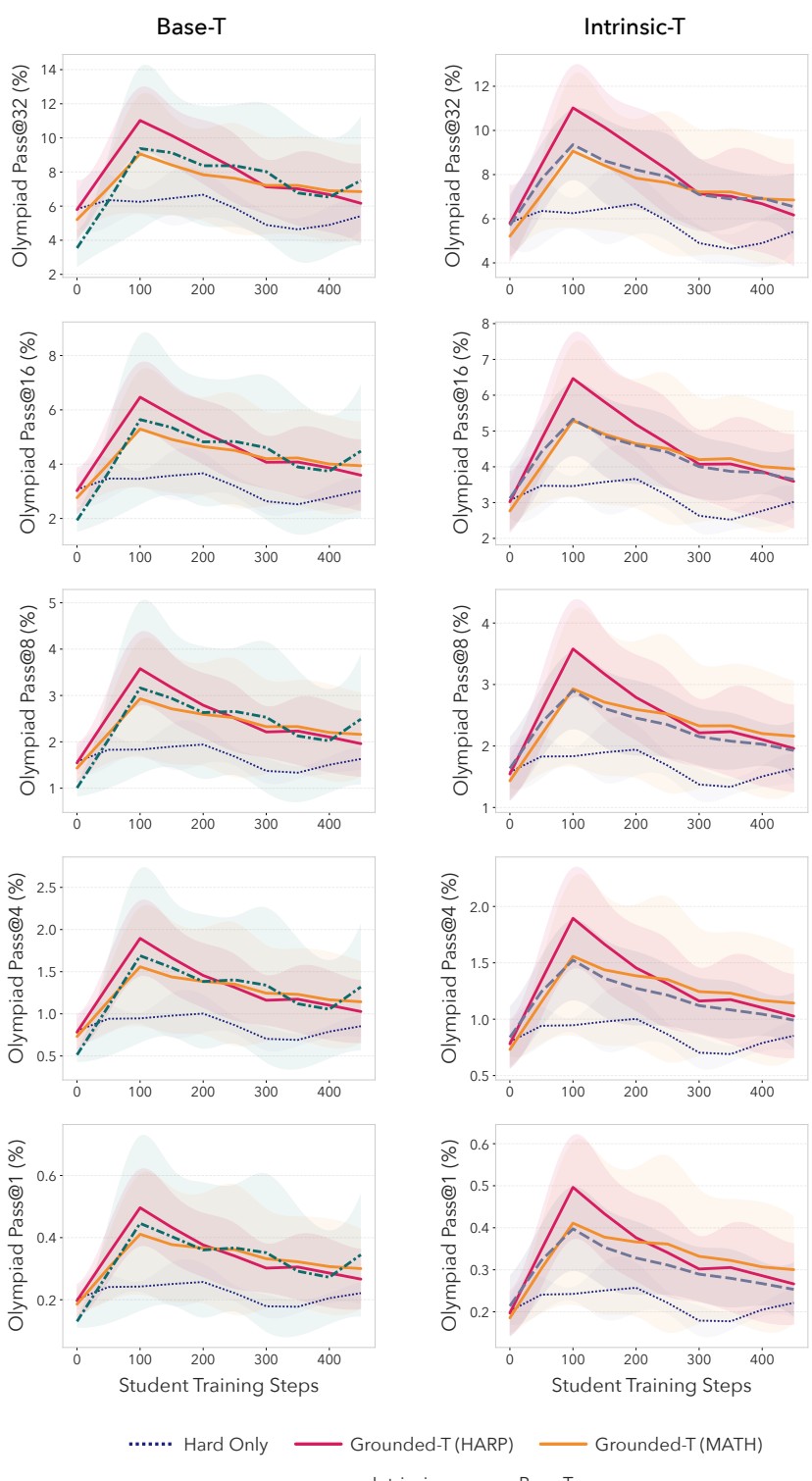

*Figure 13.* **Fail@128 test performance during student training for Olympiad with different teachers.** Each column compares training a fresh student with 128 questions from *Grounded-T* (trained with MATH and HARP) to 128 questions from a different teacher (*Hard-Only* also included for reference). Students trained with *Grounded-T* teachers have more similar mean performance to *Base-T* and *Intrinsic-T* than seen on HARP and MATH (Figures 11-12). However, *Grounded-T (HARP)* shows more stability and less variance between independent teachers than Intrinsic-T (see Figure 14).

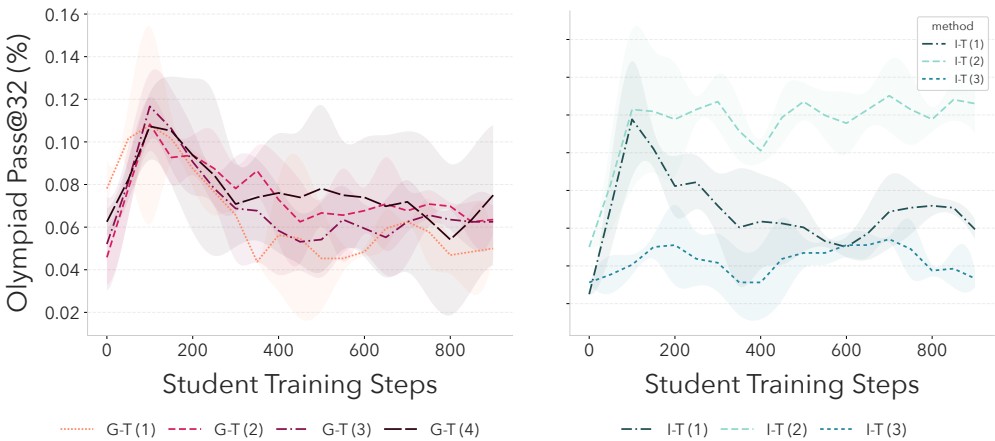

*Figure 14.* **Test Pass@32 on OlympiadBench for fresh students trained with individual *Grounded-T* teacher seeds (red) and *Intrinsic-T* teacher seeds (green).** Questions from *Grounded-T* yield consistent student trajectories on OlympiadBench across different teachers, whereas *Intrinsic-T* exhibits high variance across teachers, including a failure mode where I-T (1) causes student collapse.

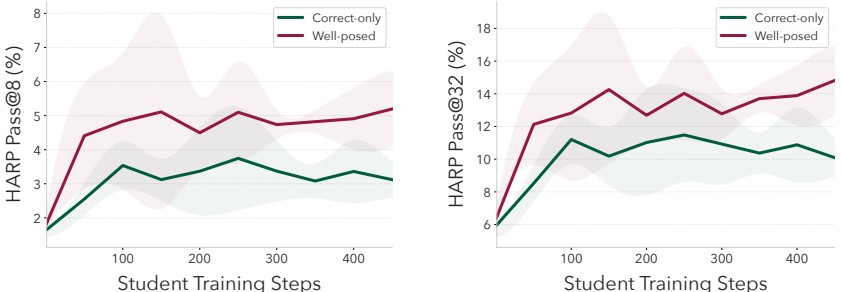

*Figure 15.* **Training on Correct-only v. Well-posed questions (HARP).** We compare training the base student on real fail@128 + *Correct-only* HARP-PQ questions, and real fail@128 + *Well-posed* HARP-PQ questions. Performance improves when adding well-posed questions with incorrect answers. Shading shows $\pm$ 1 SD over 3 seeds.

*Table 7.* **Correctness analysis and error taxonomy of synthetic questions, evaluated by `Claude-4.5-Sonnet`.** Teacher training (for both grounded and intrinsic rewards) improves the well-posedness and correctness of problems relative to the base model, with a corresponding decrease in question ambiguity errors. *Grounded-T* and PQ have fewer correct questions than *Intrinsic-T* but perform better, potentially because of greater diversity (see Table 1.)

| Category | Base | Intrinsic | Grounded | PQ |
|---|---|---|---|---|
| Well-Posed | 53.6% | 63.5% | 70.0% | 64.6% |
| Correct | 23.2% | 55.5% | 36.5% | 32.8% |
| **Error Taxonomy (% of total samples)** | | | | |
| Arithmetic Error | 23.7% | 5.7% | 29.0% | 25.0% |
| Logic Error | 5.7% | 2.3% | 6.9% | 6.5% |
| Impossibility Error | 4.7% | 2.9% | 8.2% | 4.7% |
| Ambiguity Error | 42.4% | 33.6% | 21.3% | 31.3% |
| Total Samples | 384 | 384 | 375 | 384 |

# D. Ablations

## D.1. Sampled dataset size

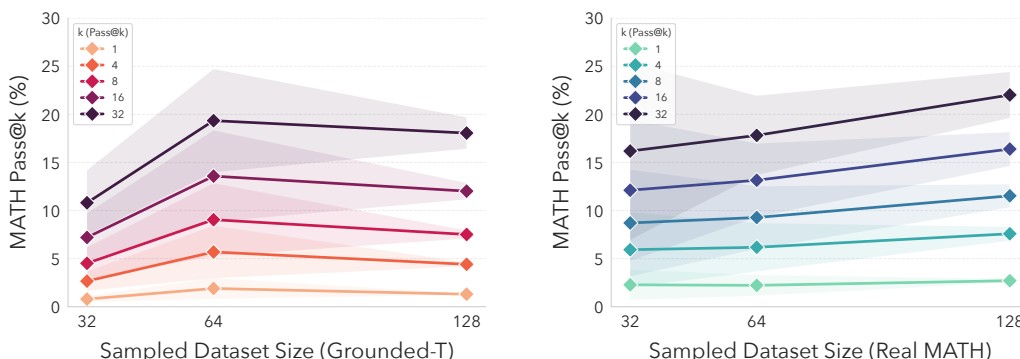

*Figure 16.* **(Left) Sampling different-sized datasets from *Grounded-T* for MATH (fail@128).** Mean and $\pm$ 1 SD across 2 teacher seeds and 2 student seeds. **(Right) Sampling different-sized datasets from the full MATH train set for MATH (fail@128).** Resampled for each seed, 3 seeds.

When training with SOAR, teacher-generated problems are partitioned into datasets that the student is trained on in the inner loop. Thus the teacher rewards are based on a specific dataset size (64 in our case). In evaluation, however, one could potentially sample any number of questions from the teacher policy. This raises the question of how the performance of sampled datasets changes with size. Is it best to sample the number of questions that the teacher was trained with, or does performance saturate at higher sampling rates?

We evaluate two teacher models trained with MATH by sampling $n \in \{32, 64, 128\}$ questions from each teacher, and training a fresh student on the sampled questions and the MATH fail@128 train set (3 seeds per run). Since teacher models are trained with $n = 64$, this covers datasets smaller, equal to, and larger than the dataset size that the teacher was trained with.

Results are shown in Figure 16 for different pass@$k$. Performance improves with increasing $n$. Sampling with 128 questions has a similar *mean* performance as sampling 64 questions but with significantly smaller *error*. This illustrates benefits (namely, consistency/reliabilty) to sampling questions from the teacher at higher rates than it was trained with. As a comparison we also perform the same experiment using *real* questions from the MATH training dataset. For all values of $n$, real MATH questions perform similarly or better, and exhibit diminishing variance with increasing numbers of questions.

## D.2. Sensitivity to Teacher Hyperparameters

We ablate $\tau$ (the teacher-reward threshold to determine if the student baseline should be promoted) and $n$ (the number of samples per dataset that teacher-generated problems are partitioned into). The teacher generates $g \cdot n$ problems per outer-RLOO iteration.

We train SOAR on MATH with $\tau \in \{0.01, 0.015\}$ and $n \in \{32, 64\}$. For each combination we train two SOAR runs for 200 steps and evaluate the final teacher checkpoints by sampling varying amounts of questions ($|\mathcal{X}| \in \{32, 64, 128\}$) and training two fresh students. Results are shown in Figure 17 for pass@8 and pass@32. Our default configuration ($n$=64, $\tau$=0.01) performs best, with $n = 64$ showing modest advantages over $n = 32$ at larger evaluation dataset sizes, which is consistent with the teacher being trained to produce larger datasets.

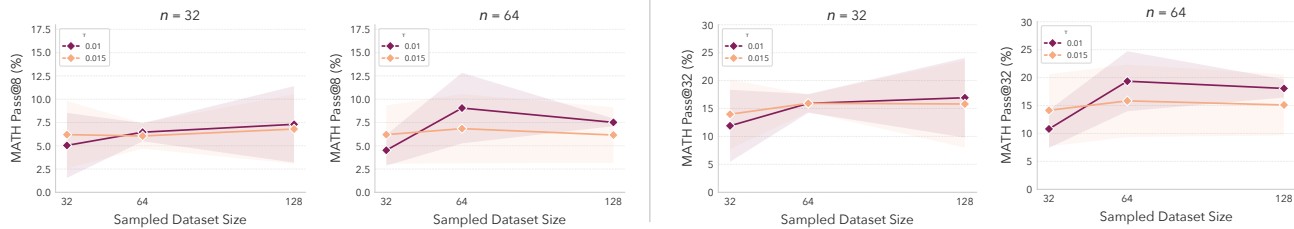

*Figure 17.* **Hyperparameter sensitivity on MATH.** We train SOAR with $\tau \in \{0.01, 0.015\}$ and $n \in \{32, 64\}$, then evaluate by training students on datasets of size $|\mathcal{X}| \in \{32, 64, 128\}$. Shaded regions indicate $\pm 1$ SD.

## D.3. Problem Generation Format.

*Table 8.* **MATH pass@$k$ (%) test accuracy on fail@128 for multi-turn teacher sampling.** We report mean and SD across four teacher seeds and 2 student seeds per teacher. Multiturn performs worse than our default single-turn setting across all pass@k and sampled dataset sizes.

| | | | | k | | |
|---|---|---|---|---|---|---|
| $n$ | $\lvert\mathcal{X}\rvert$ | **1** | **4** | **8** | **16** | **32** |
| 32 | 32 | **$0.7 \pm 0.6$** | **$2.3 \pm 1.9$** | **$4.2 \pm 3.1$** | **$7.1 \pm 4.8$** | **$11.4 \pm 6.7$** |
| | 64 | $0.5 \pm 0.3$ | $1.9 \pm 0.9$ | $3.6 \pm 1.6$ | $6.4 \pm 2.7$ | $11.0 \pm 4.0$ |
| | 128 | $0.7 \pm 0.7$ | $2.3 \pm 2.0$ | $4.0 \pm 3.3$ | $6.8 \pm 4.9$ | $11.1 \pm 7.1$ |
| 64 | 32 | $0.4 \pm 0.1$ | $1.6 \pm 0.4$ | $3.0 \pm 0.8$ | $5.2 \pm 1.4$ | $8.6 \pm 2.5$ |
| | 64 | $0.4 \pm 0.0$ | $1.5 \pm 0.2$ | $2.9 \pm 0.3$ | $5.3 \pm 0.5$ | $9.4 \pm 0.8$ |
| | 128 | $0.4 \pm 0.1$ | $1.6 \pm 0.4$ | $2.8 \pm 0.6$ | $4.8 \pm 0.9$ | $8.0 \pm 1.3$ |

In our default setup, we sample problems from the teacher by prompting it to produce a single completion that is parsed into a question/answer, and filtering out outputs that do not match the necessary format. An alternative sampling method, however, is to generate problems in separate question-answer stages (multi-turn) such that filtering is not needed:

1. Sample $\pi_\phi^T(q_i|p)$ where $p$ is a teacher prompt to generate a question.

2. Sample $\pi_\phi^T(a_i|p, q_i, p')$ where $p'$ is a prompt to generate an answer given the question.

The logprob component of the teacher RLOO loss is then $\log(\pi_\phi^T(q_i|p)) + \log(\pi_\phi^T(a_i|p, q_i, p'))$.

We execute SOAR across four seeds using this teacher-sampling formulation with our standard procedure and hyperparameters, ablating $n \in \{32, 64\}$. We observe that the teacher reward quickly plateaus and does not exceed one promotion. In Table 8 we find that across different numbers of sampled problems and values of $n$, the multi-turn sampling strategy performs worse than our default single-turn sampling.

## D.4. Model Size

We train SOAR with Llama3.1-8B-Instruct on MATH fail@128 with the same hyperparameters as those used for Llama3.2-3B-Instruct, due to the compute cost of hyperparameter tuning for meta-RL runs. With these default

hyperparameters we do not observe promotions, however find that performance still improves over the *Hard-Only* baseline, indicating that the key trend transfers to bigger models. Results are shown in Table 9, with student training curves in Figure 18.

*Table 9.* **MATH pass@k (%) test Accuracy on fail@128 with `Llama-3.1-8B-Instruct`**. Mean and SD are shown over 2 teacher seeds and 2 student seeds per teacher, at the timestep determined by training reward convergence. *Grounded-T* outperforms *Hard-Only* across all inference budgets.

| | k | | | | |
|---|---|---|---|---|---|
| **Method** | **1** | **4** | **8** | **16** | **32** |
| Base Model Inference | $0.3 \pm 0.1$ | $1.0 \pm 0.2$ | $2.0 \pm 0.4$ | $3.9 \pm 0.7$ | $7.3 \pm 1.1$ |
| *Hard-Only* | $0.4 \pm 0.1$ | $1.5 \pm 0.8$ | $2.7 \pm 1.0$ | $5.0 \pm 1.8$ | $9.1 \pm 3.2$ |
| *Grounded-T* (Ours) | $\mathbf{0.6 \pm 0.1}$ | $\mathbf{2.4 \pm 0.5}$ | $\mathbf{4.6 \pm 1.0}$ | $\mathbf{8.2 \pm 1.7}$ | $\mathbf{13.7 \pm 2.7}$ |

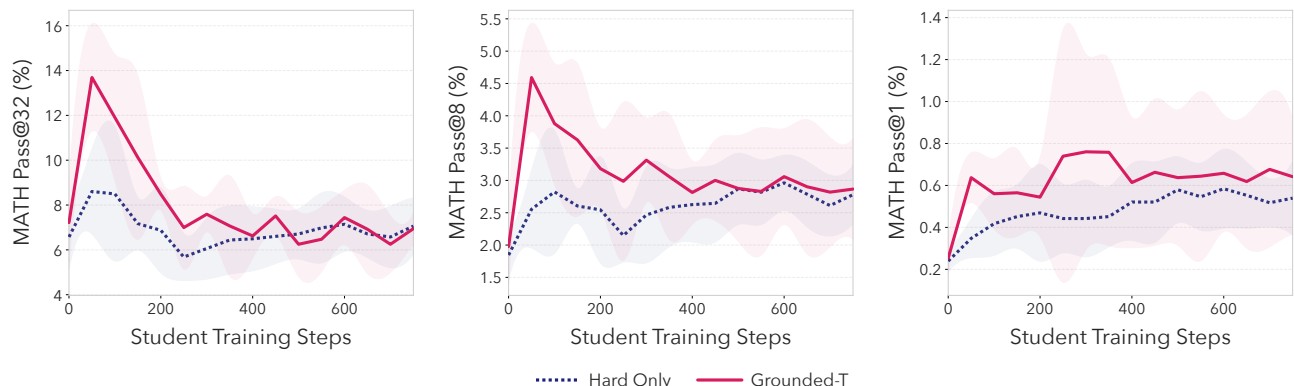

*Figure 18.* **Fail@128 test performance during student training for MATH with Llama-3.1-8B-Instruct.** We compare training a fresh student with 128 questions from *Grounded-T* to *Hard-Only*. *Grounded-T* outperforms *Hard-Only*, showing that key trends transfer to larger models. Shading shows $\pm$ 1 SD across 4 seeds (2 teacher seeds, 2 student seeds per teacher).

## E. Teacher Training Dynamics

In Figure 19 we show a representative teacher training curve for SOAR on HARP. We observe that SOAR follows a pattern of search and exploitation. The training curve exhibits periods of oscillation (search), and then a steady rise in reward from steps 18-27, culminating in a student promotion. The reward declines after the promotion, due to the improved student baseline, oscillates as the teacher adapts to the improved student, and then exhibits another rise from steps 80-86 culminating in a second promotion.

Figure 20a shows teacher training curves for *Intrinsic-T* teachers, aggregated across teacher seeds, which exhibits a smooth upward climb. Figure 20b shows that as the *Intrinsic-T* reward climbs, the diversity of teacher completions falls (diversity measured as the average pairwise cosine distance of embeddings). Meanwhile *Grounded-T* preserves the original model diversity throughout the full trajectory. This is consistent with findings in Section 5.2 (Table 1) that *Grounded-T* achieves similar question diversity to *Base-T*, whereas *Intrinsic-T* teachers collapse to a more narrow conceptual space.

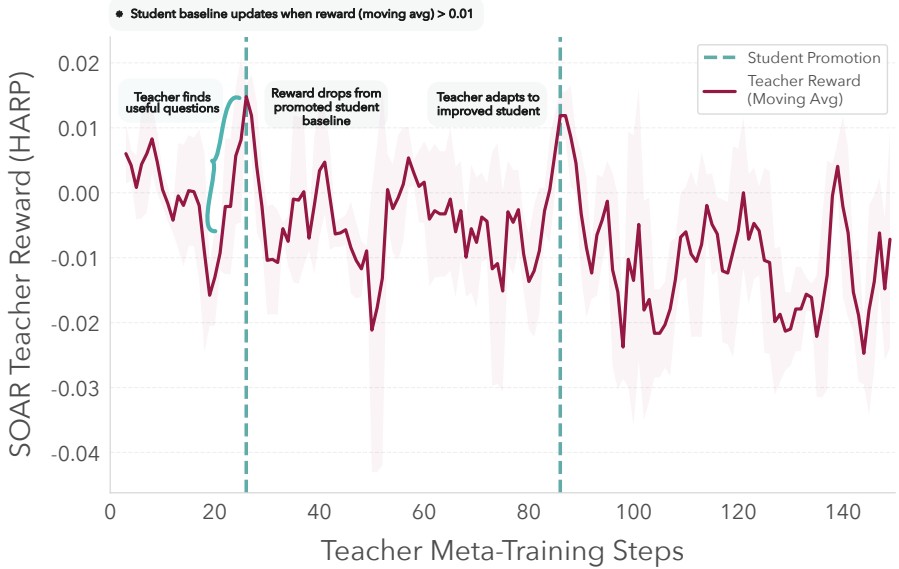

*Figure 19.* **Annotated teacher reward dynamics when training SOAR with HARP.** Shows a sample teacher trajectory from a SOAR run on HARP. The teacher follows a cyclical search-exploitation pattern. Student promotions (updating the student baseline to a trained student) are triggered when the 3-step moving average of teacher rewards exceeds $\tau = 0.01$. After each promotion, the improved student baseline makes previous curricula less useful, causing rewards to drop, and then recover as the teacher adapts and discovers questions appropriate for the improved student.

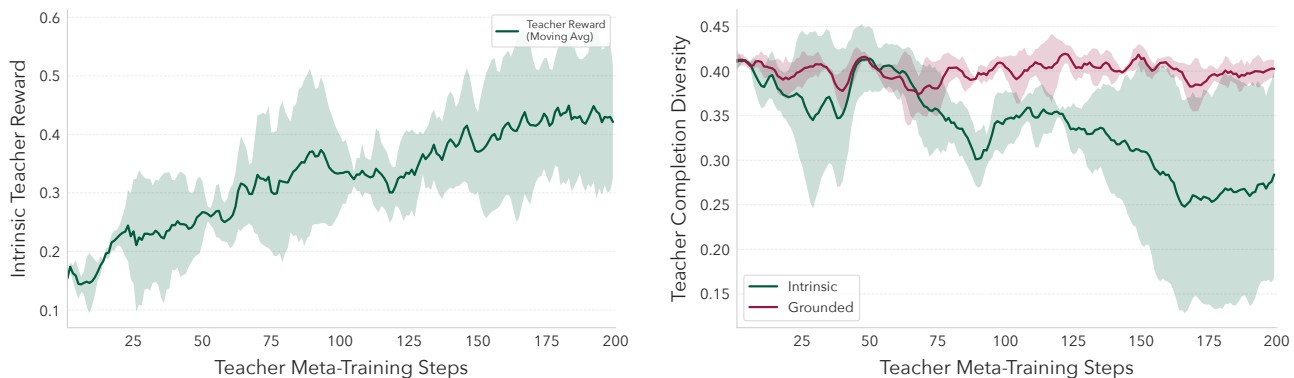

*Figure 20.* **(Left) Teacher training dynamics when training with *Intrinsic-T*.** Mean and $\pm 1$ SD over three independent training runs. **(Right) Teacher completion diversity when training with intrinsic v. grounded rewards.** Grounded rewards preserve diversity for the full run, while intrinsic teachers lose diversity as they converge. Mean and $\pm 1$ SD over three training runs for intrinsic and four for grounded (two MATH, two HARP).

