# OpenReview forum: "Teaching Models to Teach Themselves: Reasoning at the Edge of Learnability"
_ICML.cc/2026/Conference — ICML 2026 spotlight_

### Official Review · Reviewer_XbRU · 2026-03-10

**Soundness:** 3
**Presentation:** 3
**Significance:** 3
**Originality:** 3
**Overall Recommendation:** 4
**Confidence:** 3

**Summary:**

The paper introduces SOAR, a self-play bi-level meta-RL framework that enables a pretrained language model to generate its own curriculum of intermediate "stepping stone" problems. By rewarding a teacher model based on the student model's measurable progress on real, difficult datasets—rather than relying on intrinsic proxy rewards—the method successfully helps models overcome performance plateaus in sparse-reward environments. Overall, the study demonstrates that models can autonomously surface useful pedagogical signals to solve problems that initially have near-zero success rates.

**Compliance With Llm Reviewing Policy:**

Affirmed.

**Final Justification:**

The authors address my questions, so I raise my score to positive.

**Key Questions For Authors:**

See Weaknesses Part

**Limitations:**

yes

**Strengths And Weaknesses:**

Strengths:

- The paper proposes a novel self-play bi-level meta-RL framework (SOAR) designed to boost the reasoning capabilities of large language models on extremely hard mathematical problems.

- The algorithmic design is highly detailed and well-clarified throughout the text.

- The experimental results are robust and convincing, successfully demonstrating that the proposed method can kickstart learning and improve performance on difficult datasets that have initial $0/128$ success rates.

Weaknesses:

- Certain parts of the writing should be more self-contained and clear early on. For example, in the abstract, terms such as "grounded rewards" and "intrinsic rewards" are introduced somewhat vaguely. While these concepts are thoroughly detailed later in the text, defining them briefly upon their first use would improve readability.

- The experimental validation is restricted to Llama-3.2-3B-Instruct, which is a relatively older and smaller model. As the field progresses rapidly, providing results on the latest or larger-scale models is preferred to strongly support the generalizability of the claims.

- There is a lack of direct empirical comparison with other contemporary self-play methods. (e.g. some of those referred in the related work session)

---

> ### Author Rebuttal · Authors · 2026-03-31
>
> We thank the reviewer for their constructive and helpful feedback. We are glad that they found our experimental results to be robust and convincing, and our algorithm well-described. We address their comments below.
>
> > **1.** Certain parts of the writing should be more self-contained and clear early on
>
> We thank the reviewer for pointing this out. We will incorporate earlier definitions of these key terms into our revision to aid reader understanding and intuition.
>
> > **2.** The experimental validation is restricted to Llama-3.2-3B-Instruct
>
> We extend our experiments to test Llama3.1-8B-Instruct, a bigger model, on MATH fail@128 with our default hyperparameters. Due to the short time we had to tune hyperparameters, we do not observe a promotion; but nevertheless, even without promotions, we find that performance improves. We evaluate by sampling questions from the trained teacher (procedure in Section 5.2 of the paper). Performance improves over the baseline, indicating that the key trend transfers to bigger models. Results below are mean/SD over 4 seeds (2 teacher $\times$ 2 student).
>
> | Method | pass@1 | pass@4 | pass@8 | pass@16 | pass@32 |
> | :--- | :--- | :--- | :--- | :--- | :--- |
> | Base Model Inference | 0.3 ± 0.1 | 1.0 ± 0.2 | 2.0 ± 0.4 | 3.9 ± 0.7 | 7.3 ± 1.1 |
> | Hard-Only | 0.4 ± 0.1 | 1.5 ± 0.8 | 2.7 ± 1.0 | 5.0 ± 1.8 | 9.1 ± 3.2 |
> | Grounded-T (Ours) | **0.6 ± 0.1** | **2.4 ± 0.5** | **4.6 ± 1.0** | **8.2 ± 1.7** | **13.7 ± 2.7** |
>
> > **3.** There is a lack of direct empirical comparison with other contemporary self-play methods
>
> We thank the reviewer for this valuable suggestion. We extend our baselines to include SeRL [1], a contemporary self-play method that self-evolves a curriculum using an initial seed dataset, with the goal of outperforming direct RLVR on that dataset. To address the "no-verification" setting, SeRL generates question-answer pairs with majority voting and filters based on learnability. We train SeRL using MATH/HARP fail@128 as the seed sets. As Llama-3.2-3B-Instruct is a primary model used in the SeRL paper, we adopt their reported hyperparameters for these runs.
>
> Results are reported below; although SeRL training converges, it fails to match SOAR performance.
>
> | Method | MATH (pass@1) | MATH (pass@8) | MATH (pass@32) | HARP (pass@1) | HARP (pass@8) | HARP (pass@32) |
> | :--- | :---: | :---: | :---: | :---: | :---: | :---: |
> | **Hard-Only** | $0.5 \pm 0.1$ | $3.2 \pm 0.8$ | $9.6 \pm 2.6$ | $0.4 \pm 0.1$ | $2.6 \pm 0.4$ | $8.2 \pm 1.0$ |
> | **SOAR-PS** | $1.0 \pm 0.2$ | $6.8 \pm 1.1$ | $18.1 \pm 2.4$ | $0.6 \pm 0.1$ | $3.9 \pm 0.6$ | $11.8 \pm 1.2$ |
> | **SOAR-PQ** | $\mathbf{1.7 \pm 1.0}$ | **$\mathbf{8.5 \pm 3.7}$** | **$\mathbf{18.9 \pm 5.3}$** | **$\mathbf{0.7 \pm 0.3}$** | **$\mathbf{4.5 \pm 1.3}$** | **$\mathbf{12.3 \pm 2.0}$** |
> | **SeRL (step 50)** | $0.5 \pm 0.0$ | $3.9 \pm 0.0$ | $12.3 \pm 0.5$ | $0.4 \pm 0.0$ | $3.0 \pm 0.5$ | $10.7 \pm 1.5$ |
> | **SeRL (step 100)** | $0.5 \pm 0.0$ | $3.7 \pm 0.3$ | $12.2 \pm 0.5$ |$0.4 \pm 0.0$ | $3.5 \pm 0.1$ | $10.8 \pm 0.3$ |

---

> > ### Author Rebuttal · Reviewer_XbRU · 2026-04-05
> >
> > My concerns have been adequately addressed.

---

### Official Review · Reviewer_8hoM · 2026-03-12

**Soundness:** 3
**Presentation:** 2
**Significance:** 3
**Originality:** 3
**Overall Recommendation:** 4
**Confidence:** 3

**Summary:**

This paper studies whether a model can improve on problems it cannot initially solve by generating its own stepping-stone curriculum. The authors propose SOAR, a teacher-student meta-RL framework in which a teacher generates synthetic question-answer pairs and is rewarded according to the student’s improvement on hard fail@128 subsets. Experiments on MATH, HARP, and held-out OlympiadBench show that the generated questions outperform direct training on the hard sets, that grounded teacher rewards are more stable than intrinsic ones, and that question structure seems to matter more than answer correctness in this regime.

**Compliance With Llm Reviewing Policy:**

Affirmed.

**Key Questions For Authors:**

Please see weaknesses.

**Limitations:**

yes

**Strengths And Weaknesses:**

Strengths

1. The setting is well motivated: standard RLVR can stall completely when the initial success rate is essentially zero, and the paper gives a concrete way to study that failure mode rather than avoiding it. Using a teacher to generate synthetic questions is not new by itself, but grounding the teacher reward in measured student improvement on real hard problems is a strong design choice, and the bilevel teacher-student setup is clearly presented.

2. The paper includes multi-seed results, comparisons against direct hard-only training and intrinsic-reward teachers, transfer to OlympiadBench, and useful analysis of teacher stability and diversity. I also found the result about question structure mattering more than answer correctness quite interesting.

Weaknesses

1. My main concern is generality. All experiments are done with a single model, Llama-3.2-3B-Instruct, and all of them are in math reasoning. That makes the paper feel more like a strong proof of concept than a broadly validated method. It is not yet clear whether the same bilevel loop will remain effective at larger scales or in other domains.

2. No experiments are conducted on non-math domains (e.g., coding, science QA), so domain generalizability is unverified.

3. The bilevel optimization in Eq. 1 is stated as a formal objective, but the actual optimization procedure (nested RLOO) is only a heuristic approximation whose convergence properties are not analyzed.

---

> ### Author Rebuttal · Authors · 2026-03-31
>
> We thank the reviewer for their insightful feedback. We are glad that they found the setting well-motivated, and highlighted our design choices, multi-seed results and empirical findings. We address their comments below.
>
> > **1.** All experiments are done with a single model, Llama-3.2-3B-Instruct, and all of them are in math reasoning...domain generalizability is unverified.
>
>
> We extend our experiments to test generality along two axes:
> 1. **Model size**: We train with Llama3.1-8B-Instruct on MATH fail@128 with our default hyperparameters. Due to the short time we had to tune hyperparameters, we do not observe a promotion; but nevertheless, even without promotions, we find that performance improves. We evaluate by sampling questions from the trained teacher (procedure in Section 5.2 of the paper). Performance improves over the baseline, indicating that the key trend transfers to bigger models. Results below are mean/SD over 4 seeds (2 teacher $\times$ 2 student).
>
> | Method | pass@1 | pass@4 | pass@8 | pass@16 | pass@32 |
> | :--- | :--- | :--- | :--- | :--- | :--- |
> | Base Model Inference | 0.3 ± 0.1 | 1.0 ± 0.2 | 2.0 ± 0.4 | 3.9 ± 0.7 | 7.3 ± 1.1 |
> | Hard-Only | 0.4 ± 0.1 | 1.5 ± 0.8 | 2.7 ± 1.0 | 5.0 ± 1.8 | 9.1 ± 3.2 |
> | Grounded-T (Ours) | **0.6 ± 0.1** | **2.4 ± 0.5** | **4.6 ± 1.0** | **8.2 ± 1.7** | **13.7 ± 2.7** |
>
>
> 2. **Domain:** We evaluate Llama3.2-3B-Instruct on the Knights & Knaves (K&K) dataset [1] containing verifiable logic puzzles. We chose this dataset based on its use by prior work to test RLVR for logic reasoning [2], and to evaluate structurally different reasoning paths from quantitative datasets.
>
>     Unlike our other benchmarks, we find that Hard-Only training on fail@128 improves performance substantially due to format being a greater bottleneck in this domain than for math; thus the partial format reward has more effect (0.4% -> 20.1% pass@1). Thus this is outside the "zero-reward" regime conducive to improvement via SOAR. We are actively working on a different dataset but are unable to deliver by the rebuttal deadline and hope to respond with more results during the discussion period.
>
> > **2.** The bilevel optimization in Eq. 1 is stated as a formal objective, but the actual optimization procedure (nested RLOO) is only a heuristic approximation whose convergence properties are not analyzed.
>
> The reviewer is correct that our implementation via nested RLOO is a heuristic approximation of the formal bilevel objective. We do so because the exact objective is computationally intractable, requiring "backpropagation through gradient descent" [3-4].
> While a formal convergence analysis of nested RL for LLMs remains an important open challenge, we justify our approximation empirically by showing that teachers trained with different seeds produce similar student trajectories (Section 5.2; Figure 5). We also include an annotated example of a converging teacher curve (Figure 16).
>
> We will add clarification of this distinction in our revision.
>
> [1] X., Chulin et al. “On Memorization of Large Language Models in Logical Reasoning.” IJCNLP-AACL (2024)
> [2] S., Zafir et al. “REASONING GYM: Reasoning Environments for Reinforcement Learning with Verifiable Rewards.” NeurIPS 2025.
> [3] T. Wang et al. Dataset distillation. arXiv preprint
> arXiv:1811.10959, 2018.
> [4] Y. Feng et al. Embarrassingly simple dataset distillation. ICLR, 2024.

---

> > ### Author Rebuttal · Reviewer_8hoM · 2026-04-03
> >
> > Thanks for the rebuttal. I maintain my positive assessment.

---

> > > ### Author Response · Authors · 2026-04-07
> > >
> > > We thank the reviewer for clarifying that their concerns are fully addressed, and for maintaining their positive score.
> > >
> > > Following the reviewer's earlier remark on non-math domains, we have run a limited version of SOAR (due to time constraints) on a physics dataset. We use the physics subset of the WebInstruct-Verified dataset [1], including topics such as optics, astronomy, fluids, and thermodynamics. We train SOAR with 80 outer-loop steps and $r=2$, and evaluate the generated PQ questions following the same procedure as Section 5.1 of the paper. Performance consistently improves over the Hard-Only baseline, indicating that our key trends can transfer to non-math datasets.
> > >
> > > | Method | pass@1 | pass@4 | pass@8 | pass@16 | pass@32 |
> > > | :--- | :--- | :--- | :--- | :--- | :--- |
> > > | Base Model Inference | 0.2± 0.0 | 0.9 ± 0.2 | 1.7 ± 0.4 | 3.4 ± 0.7 | 6.6 ± 1.4 |
> > > | Hard-Only | 0.3 ± 0.1 | 1.4 ± 0.3 | 2.8 ± 0.5 | 5.2 ± 0.8 | 9.6 ± 1.3 |
> > > | SOAR-PQ (Ours) | **0.5 ± 0.1** | **2.0 ± 0.5** | **3.8 ± 0.8** | **7.0 ± 1.0** | **12.1 ± 0.9** |
> > >
> > > [1] Ma, Xueguang et al. "General-Reasoner: Advancing LLM Reasoning Across All Domains". NeurIPS, 2025.

---

### Official Review · Reviewer_Qutw · 2026-03-12

**Soundness:** 2
**Presentation:** 3
**Significance:** 2
**Originality:** 3
**Overall Recommendation:** 4
**Confidence:** 3

**Summary:**

This paper studies a failure regime of RL with verifiable rewards (RLVR) for reasoning: when the initial success rate on a target dataset is essentially zero, direct RL fine-tuning provides no learning signal and stalls. The authors propose SOAR (Self-Optimization via Asymmetric RL), a teacher–student meta-RL framework where a teacher model generates synthetic (question, answer) pairs for a student to train on, and the teacher is rewarded by the student’s measured improvement on a held-out subset of the real hard problems. The paper also claims that question structure/well-posedness matters more than answer correctness.

**Compliance With Llm Reviewing Policy:**

Affirmed.

**Final Justification:**

The authors’ rebuttal addressed my 2 major concerns about the contribution of incorrect answers under RLVR and the compute-performance trade-off. Their replies are well-written with abundant and detailed experiment results, and my questions are all answered. There is also no logic error in their work and rebuttal. So, I decided to raise the point to 4.

**Key Questions For Authors:**

1. Generalization across reward-question sampling: How sensitive is teacher learning to the QR subsampling scheme (size, difficulty mix, fixed vs resampled, different random seeds)? Can you show that teachers trained under one QR sampling distribution improve on a disjoint QR distribution and still yield strong PQ performance?

2. Why do incorrect answers help under RLVR? Provide a deeper analysis of the student’s learning signal when trained on teacher QA with low correctness. For example: does the student’s reward improvement come from improved exploration, improved formatting/chain-of-thought style, or learning transferable subskills? Can you stratify PQ by correctness/well-posedness and show which bins drive gains?

3. Compute–performance tradeoff: Provide an explicit compute accounting: total student-update steps (including parallel runs r), total tokens, wall-clock estimates, and compare against “equivalent compute” alternatives (e.g., more Hard-Only steps, larger groups, or longer training). The current “group size 128” ablation is a start but not a full accounting.

4. Robustness beyond 3B and beyond math: Do the key trends hold for other model sizes (smaller and larger) and at least one non-math verifiable domain? If not feasible, clarify the expected barriers (reward noise, policy collapse, sampling budget).

**Limitations:**

High computational overhead due to nested RL loops and parallel student trainings for variance reduction; practicality at scale is uncertain.

Potential overfitting to the grounded reward set (QR sampled from Dtrain) without stronger tests of robustness to reward sampling changes.

Limited mechanistic understanding of how learning proceeds when synthetic answers are frequently incorrect, and whether this induces long-term brittleness or shortcut learning.

**Strengths And Weaknesses:**

**Strengths**

Targets a real and important failure mode. The “fail@128 / 0/128 success” setting is a crisp operationalization of the sparse-reward plateau where naive RLVR does not progress, and the paper directly attacks this regime.

Grounded outer-loop signal is conceptually clean. Rewarding the teacher by student improvement on real hard problems (ACC(π′(QR)) − ACC(π(QR))) is a sensible antidote to reward hacking/drifting that can occur with intrinsic/proxy rewards in self-play, and the paper provides comparative evidence versus an intrinsic learnability baseline.

Empirical hygiene is better than typical for bilevel/self-play papers. The authors report multi-seed mean±SD in main plots, and explicitly compare against “more compute to Hard-Only” (e.g., larger group size) to argue gains are not just extra RL budget.

Interesting observation about correctness vs utility. The finding that many useful teacher-generated questions have incorrect answers, while still improving the student, is provocative and—if robust—could influence how the community thinks about “synthetic data quality.”


**Weaknesses**

Potential reward leakage/evaluation coupling. The teacher is optimized on improvement measured on a subset QR sampled from Dtrain, and later PQ is evaluated by training fresh students on PQ mixed with fail@128 train and testing on Dtest. The design is reasonable, but the paper needs stronger evidence that the outer-loop reward is not overfitting to artifacts of QR sampling (e.g., repeatedly “teaching to the QR distribution”) and that gains generalize across different QR sampling schemes / splits.


The compute cost and scalability remain unclear. The approach requires many inner-loop student trainings per outer step (and averaging over r parallel trainings to reduce variance). The paper frames the work as proof-of-concept and notes compute as a primary limitation, but the current evidence is mostly at 3B scale with 200 outer steps; it is not yet clear whether the method is practical at the scales where reasoning RL is typically deployed.


Synthetic QA correctness is low; mechanism is under-explained. Reporting that only ~33% of PQ have fully correct solutions while ~63% are well-posed suggests the student may be learning from structure/format/latent concepts rather than supervised correctness. This is interesting, but it also raises concerns about how incorrect answers help under RLVR and whether the improvement is stable or fragile (e.g., risk of learning spurious heuristics). The paper’s causal story here needs deeper mechanistic/diagnostic support.


Baselines could be stronger / broader. The comparison to an intrinsic learnability teacher is helpful, but the related-work space includes other self-evolving curriculum / self-training setups; the paper would benefit from either (i) stronger implementation-aligned baselines, or (ii) clearer justification that those baselines are not applicable in this “no verification” math setting.

---

> ### Author Rebuttal · Authors · 2026-03-31
>
> We thank the reviewer for their insightful comments and positive assessment of our framework's design, empirical hygiene, and findings. We address questions below.
>
> > **1.** How sensitive is teacher learning to the QR subsampling scheme?
>
> The size of the subsampled $QR \sim D_{train}$ at each teacher step can have effect. If $QR$ is too small, then spurious improvements in a few questions cause large reward swings. We fix $|QR|=64$ to balance reward stability with evaluation cost.
>
> Our experiments also ablate the subsampling scheme:
>
> -  **Repetition of QR questions over teacher steps**: We implicitly ablate the amount of overlap between subsampled $QR$'s through our choice of datasets. HARP and MATH have different-sized $D_{train}$ pools (359 questions for MATH, 714 for HARP), thus $QR$ questions are repeated more often for MATH. SOAR learns effectively in both cases.
> -  **Random seed**: For each dataset we train 4 teachers; teachers trained with different seeds produce similar student performance (Section 5.2).
>
> > **2.** Do teachers trained under one QR sampling distribution improve on a disjoint QR distribution?
>
> We show this through **our OlympiadBench evaluations (Fig. 4; Sec. 5.1)**. PQ-HARP and PQ-MATH each improve student performance on OlympiadBench. In this experiment, OlympiadBench is a disjoint distribution from $D_{train}$ and $QR$.
>
> > **3.** Baselines could be stronger / broader
>
> We **extend our baselines to include SeRL [1], a contemporary self-play method** that self-evolves a curriculum using a seed dataset. For the "no-verification" setting, SeRL generates question-answer pairs with majority voting and filters with learnability. We train SeRL with MATH/HARP fail@128 as the seed sets, and adopt their reported Llama-3.2-3B-Instruct hyperparameters. **Although SeRL training converges, it performs worse than SOAR. Our full results table is in section 3 of our response to Reviewer XbRU.**
>
> > **4.** Why do incorrect answers help under RLVR?
>
> Following the reviewer's suggestion, **we run a controlled experiment to isolate the effects of adding well-posed questions with incorrect answers**. We compare training on the subset of PQ-HARP questions that have correct answers (*fully-correct*), and the subset that have well-posed questions (*well-posed*). Note that *fully-correct* is strictly a subset of *well-posed*.
>
> **For a detailed overview of our findings, please refer to Section 2 of our response to Reviewer UGt8**. To summarize here:
> - *Well-posed* yields better performance than *fully-correct* ([**link to pass@32 plot**](https://imgur.com/a/vYOFzvh); [**link to pass@8 plot**](https://imgur.com/mkLazRO)).
> - Token-level entropy and rollout diversity decline faster with *well-posed* training ([**link to entropy plot**](https://imgur.com/uFeNNK0); [**link to rollout diversity plot**](https://imgur.com/pIbMY6q)), indicating that the model converges faster to a specific style/template of reasoning.
> - *Well-posed* demonstrates more stable initial training dynamics (lower initial grad norms: 1448 $\pm$ 708 v. 3008 $\pm$ 507).
> - In all cases (Hard-Only, *fully-correct*, *well-posed*) format match reaches 100% within the first ~30 steps; thus, formatting is not the main driver of performance gains.
>
> Regarding the risk of 'spurious heuristics,' our transfer results on OlympiadBench (Figure 4) suggest that our synthetic questions impart more general reasoning paths beyond dataset-specific shortcuts.
>
> > **5.** Compute–performance tradeoff
>
> **Accounting:**
> - Wall-clock estimates are in Appendix B.9.
> - The number of student update steps during Meta-RL is:
> (10 steps) $\times$ 4 ($r$) $\times$ 4 ($g$) $\times$ 200 (outer steps) = 32000 total.
> In practice, promotions often occur in the first 100 steps (Fig. 16), so most PQ datasets reflect a smaller number of update steps.
>
> **We add another increased-compute baseline that trains Hard-Only for more steps**. Due to time constraints we are unable to provide a compute-equivalent run (16k-32k steps). However, we extend training from 2k steps (original baseline) to 6.5k steps. **Performance *decreases* when extending the run**, indicating that additional compute during GRPO is unable to unlock the learning we achieve through our teacher-student setup ([**link to plot**](https://imgur.com/a/Q26DGZn)).
>
> Our work provides a proof of concept of grounded rewards for expanding a model's learnability frontier. Efficiency is indeed important for deployment, and a critical avenue for future work; for instance, incorporating diversity/repetition heuristics may reduce training time [1-2]
>
> > **6.** Robustness beyond 3B and beyond math
>
> We show that **our key trends extend to a larger model, Llama3.1-8B-Instruct**, and initial exploration of other domains. Please refer to **Section 1 of our response to Reviewer 8hoM** for details/results.
>
> [1] Fang, Wenkai, et al. "Serl: Self-play reinforcement learning...", 2025.
>
> [2] Huang, Chengsong, et al. "R-zero...", 2025.

---

> > ### Author Rebuttal · Reviewer_Qutw · 2026-04-02
> >
> > Thank you for your response. I have decided to increase my score by one point.

---

> > > ### Author Response · Authors · 2026-04-07
> > >
> > > We thank the reviewer for clarifying that their concerns are fully addressed, and for increasing their score.
> > >
> > > Following the reviewer's earlier request for a non-math domain, we have run a limited version of SOAR (due to time constraints) on a physics dataset. We use the physics subset of the WebInstruct-Verified dataset [1], including topics such as optics, astronomy, fluids, and thermodynamics. We train SOAR with 80 outer-loop steps and $r=2$, and evaluate the generated PQ questions following the same procedure as Section 5.1 of the paper. Performance consistently improves over the Hard-Only baseline, indicating that our key trends can transfer to non-math datasets.
> > >
> > > | Method | pass@1 | pass@4 | pass@8 | pass@16 | pass@32 |
> > > | :--- | :--- | :--- | :--- | :--- | :--- |
> > > | Base Model Inference | 0.2± 0.0 | 0.9 ± 0.2 | 1.7 ± 0.4 | 3.4 ± 0.7 | 6.6 ± 1.4 |
> > > | Hard-Only | 0.3 ± 0.1 | 1.4 ± 0.3 | 2.8 ± 0.5 | 5.2 ± 0.8 | 9.6 ± 1.3 |
> > > | SOAR-PQ (Ours) | **0.5 ± 0.1** | **2.0 ± 0.5** | **3.8 ± 0.8** | **7.0 ± 1.0** | **12.1 ± 0.9** |
> > >
> > > [1] Ma, Xueguang et al. "General-Reasoner: Advancing LLM Reasoning Across All Domains". NeurIPS, 2025.

---

### Official Review · Reviewer_UGt8 · 2026-03-12

**Soundness:** 4
**Presentation:** 3
**Significance:** 3
**Originality:** 3
**Overall Recommendation:** 5
**Confidence:** 4

**Summary:**

This paper introduces a meta-RL framework for overcoming sparse-reward plateaus in RL training. A bi-level RL loop trains a teacher model to propose questions and a student model to solve them, with the aim of overcoming the plateau through signals from the proposed questions. The core novelty of this work compared to previous self-play methods is that the teacher is trained using a grounded, rather than intrinsic, signal. This means that the teacher is rewarded when its proposed questions lead to student generalization to the target data, and not simply when the proposed questions are of appropriate difficulty for the student.

In the outer loop, the teacher proposes G sets of questions and answers. Parallel students are trained on each set and evaluated on the target data. The teacher is rewarded based on the resulting improvement from the proposed dataset. In the inner loop, the student is trained on a mix of synthetic and real questions as usual.

They study a setting where the training data is very difficult for the base model (fail@128) so that naive RL training is very sample-inefficient due to lack of training signal. Training with their method outperforms naive RL training (Hard only) and intrinsic teacher rewards (Intrinsic-T). Analysis shows that the generated questions themselves are valuable – training a fresh student on them outperforms the best student from the meta-RL training (PQ > PS).

Additional analyses focuses on understanding design decisions around the teacher model. It is shown that meta-RL sharpens teacher question distribution to pedagogical subsets (i.e., good for training student model). Moreover, teacher models trained on intrinsic proxies are unstable and sometimes collapse, indicating reward hacking. Analysis of diversity also shows that the grounded teacher learns more diverse questions than the intrinsic teacher.

**Compliance With Llm Reviewing Policy:**

Affirmed.

**Final Justification:**

See review and rebuttal acknowledgement. My initial assessment was already positive, and the authors' response helped clarify my remaining questions.

**Key Questions For Authors:**

- The paper studies the setting where essentially no "stepping stone" data is provided. Their evaluations also show that using the actual training set is more effective than relying entirely on teacher-generated data. If allowed to use the full training set, is the method still able to complement existing data to improve performance?

- A premise in the experiments is that the training data is hard enough that the model gets essentially no signal (fail@128). But this also means that the teacher is unlikely to get signal, since it only gets reward when the student gets non-zero performance on the training data. Intrinsic rewards seem to get around this since they only need to propose questions that are reasonable for the student to solve. Could intrinsic and grounded signals be complementary in this way?

**Limitations:**

yes

**Strengths And Weaknesses:**

Strengths:
The paper proposes a novel framework for self-play RL and shows promising empirical results, with extensive ablation and analyses.
I find the experiments to be well-designed. The paper is also generally well-written.
The paper presents a novel conceptual shift from existing self-play work, which as far as I know have been focused on Intrinsic supervision for the teacher.

Weaknesses:
- I am not fully convinced on section 5.3. The paper presents evidence that Grounded-T outperforms Instrinsic-T despite lower answer correctness, and argues that question structure itself can provide useful gradient signal, but does not propose an underlying mechanism. How does reinforcing inaccurate answers on well-posed questions ultimately improve student ability to solve real questions?

---

> ### Author Rebuttal · Authors · 2026-03-31
>
> We thank the reviewer for the positive assessment of our work. We appreciate that they found paper to be conceptually novel, and the experiments/analysis to be well-designed and extensive.
>
> We have addressed remaining comments and questions below.
>
> > **1.** How does reinforcing inaccurate answers on well-posed questions ultimately improve student ability to solve real questions?
>
> We appreciate the reviewer's question about how well-posed questions with incorrect answers can aid learning. **To address this, we run a controlled experiment to isolate the effects of adding well-posed questions with incorrect answers.** We compare training on the subset of PQ-HARP questions that have correct answers (*fully-correct*), and the subset that have well-posed questions (*well-posed*). Note that *fully-correct* is strictly a subset of *well-posed*. We observe:
>
> - **Performance improvement:** Training on a combination of *well-posed* and fail@128 train performs better than *fully-correct* and fail@128 train, showing concretely that there are performance gains from adding well-posed questions with incorrect answers. ([**link to pass@32 plot**](https://imgur.com/67AdEFQ); [**link to pass@8 plot**](https://imgur.com/mkLazRO)).
> - **Directed search**: Token-level entropy and rollout diversity decline faster with *well-posed* training ([**link to entropy plot**](https://imgur.com/uFeNNK0); [**link to rollout diversity plot**](https://imgur.com/pIbMY6q)). This indicates that the model converges faster to a specific style or template of reasoning and logic flow that makes harder problems a shorter "jump" later in training.
> - **Stability**: *well-posed* has lower initial gradient norms than *fully-correct* (1448 $\pm$ 708 v. 3008 $\pm$ 507). The dataset with some incorrect answers produces weaker or more diffuse reward signals, resulting in smaller early policy updates.
> - **Performance decoupled from format**: In all cases (Hard-Only, *fully-correct*, *well-posed*) the format match reaches 100% within the first ~30 steps; thus, formatting is not the main driver of performance gains.
>
> We gain additional insight from the taxonomy of error types in Table 7 of the paper. PQ and Grounded-T questions have fewer ambiguity errors, and a higher proportion of arithmetic errors than the base teacher. Such questions may still reinforce conceptually relevant chains-of-thought (even if leading to an arithmetically incorrect answer) that are useful for initial training stages.
>
> > **2.** If allowed to use the full training set, is the method still able to complement existing data to improve performance?
>
> As the reviewer notes, real, curated intermediate datasets generally perform better than synthetic stepping stones. We note though that for HARP, PQ outperforms an equivalent number of real intermediate HARP or MATH training problems (Table 5), indicating that it may be complementary. To extend our method to a full dataset already containing stepping stones, one could incorporate further reward terms such as repetition penalties to ensure that the synthetic data targets gaps in the existing dataset.
>
> > **3.** Could intrinsic and grounded signals be complementary in this way?
>
> We thank the reviewer for the interesting suggestion.
>
> A key finding of our work is that *in the settings that we test, it is easier to sample tasks that induce non-zero signal on hard problems, than to directly sample correct answers to those hard problems*. As the reviewer points out, it is likely that there are tasks sufficiently difficult that this is not the case.
>
> In preliminary experiments we attempted to use learnability, an intrinsic reward, as a "warm-start", however found that this reduced the teacher's entropy such that there was insufficient exploration in the grounded-reward phase. Moreover, this learnability warm-start led to reward-hacking, as we observe in the paper. This is consistent with findings in prior work that intrinsic rewards are prone to entropy collapse and reward-hacking [1]. However, it is possible that the two can be complementary with careful tuning; this is a valuable avenue for future work.
>
> [1] Justin Yang Chae, Md Tanvirul Alam, and Nidhi Rastogi. Towards understanding self-play for llm reasoning, 2025.

---

> > ### Author Rebuttal · Reviewer_UGt8 · 2026-04-02
> >
> > Thank you for your response. I maintain my positive assessment.

---

### Decision · Program_Chairs · 2026-04-30

**Decision:**

Accept (spotlight)

**Comment:**

The paper proposes SOAR to address a key challenge in RLVR: when a model’s pass rate on hard problems is near zero, standard RL yields no useful training signal since all samples receive zero reward. The method introduces a teacher–student loop, where a teacher model generates synthetic questions for the student, and is rewarded based on the student’s improvement on held-out hard problems. This approach leads to consistent gains on difficult reasoning tasks.

All four reviewers reached consensus in their evaluations. The rebuttal addressed a key concern with a controlled ablation and included an analysis showing that extended hard-only training degrades performance, ruling out the possibility that the teacher loop is merely a compute surrogate. For the camera-ready paper, we also suggest adding comparisons to other self-play/bootstrapping approaches designed to handle zero-reward settings. Overall, reviewers praised the novelty of the reward design and the rigor of the empirical analysis. Thus, we recommend acceptance of the paper.